# Approximately Equivariant Recurrent Generative Models for Quasi-Periodic Time Series with a Progressive Training Scheme

**Ruwen Fulek**                                                                  *ruwen.fulek@th-owl.de*
*Institut für industrielle Informationstechnik*
*Technische Hochschule Ostwestfalen-Lippe*
*32657 Lemgo*

**Markus Lange-Hegermann**                                 *markus.lange-hegermann@th-owl.de*
*Institut für industrielle Informationstechnik*
*Technische Hochschule Ostwestfalen-Lippe*
*32657 Lemgo*

**Reviewed on OpenReview:** *https://openreview.net/forum?id=KHk5EECG3Z*

## Abstract

We present a simple yet effective generative model for time series, based on a Recurrent Variational Autoencoder that we refer to as AEQ-RVAE-ST. Recurrent layers often struggle with unstable optimization and poor convergence when modeling long sequences. To address these limitations, we introduce a training scheme that subsequently increases the sequence length, stabilizing optimization and enabling consistent learning over extended horizons. By composing known components into a recurrent, approximately time-shift-equivariant topology, our model introduces an inductive bias that aligns with the structure of quasi-periodic and nearly stationary time series. Across several benchmark datasets, AEQ-RVAE-ST matches or surpasses state-of-the-art generative models, particularly on quasi-periodic data, while remaining competitive on more irregular signals. Performance is evaluated through ELBO, Fréchet Distance, discriminative metrics, and visualizations of the learned latent embeddings.

## 1 Introduction

Time series data, particularly sensor data, plays a crucial role in science, industry, energy, and health. With the increasing digitization of companies and other institutions, the demand for advanced methods to handle and analyze time series sensor data continues to grow. Sensor data often exhibits distinct characteristics: it is frequently multivariate, capturing several measurements simultaneously, and may involve high temporal resolutions, where certain anomalies or patterns of interest only become detectable in sufficiently long sequences. Furthermore, such data commonly displays quasi-periodic behavior, reflecting repetitive patterns influenced by the underlying processes. For generative models, this raises the challenge of how to embed inductive biases that emphasize relative temporal dynamics over absolute time, encouraging the model to treat repeating structures consistently regardless of their position in the sequence. These unique properties present both opportunities and challenges in the development of methods for efficient data synthesis and analysis, which are essential for a wide range of applications.

Throughout this paper, we use the term *quasi-periodic* in the applied anomaly-detection sense, referring to time series that exhibit repetitive motifs recurring with imperfect regularity due to variable cycle lengths, phase jitter, drift, noise, missing cycles, or occasional anomalies. This usage is consistent with the anomaly-detection and time-series segmentation literature (Yang et al., 2025; Liu et al., 2022; Zangrando et al., 2022;

Tang et al., 2023) and differs from the classical mathematical notion of quasi-periodicity, which typically refers to structured signals generated by a finite number of incommensurate frequencies.

Time series data analysis spans tasks such as forecasting (Siami-Namini et al., 2019), imputation (Tashiro et al., 2021; Luo et al., 2018), anomaly detection (Hammerbacher et al., 2021), and data generation. Of these, data generation stands out as the most general task, as advances in generative methods often yield improvements across the entire spectrum of time series applications (Murphy, 2022).

Recurrent neural networks, particularly Long Short-Term Memory (LSTM) networks (Hochreiter & Schmidhuber, 1997), are well-known for their ability to model temporal dynamics and capture dependencies in sequential data. However, their effectiveness tends to diminish with increasing sequence length, as maintaining long-term dependencies can become challenging (Zhu et al., 2023) where in contrast, convolutional neural networks (CNNs) (LeCun et al., 1998) demonstrate superior scalability for longer sequences (Bai et al., 2018). For instance, TimeGAN (Yoon et al., 2019) represents a state-of-the-art approach for generating synthetic time series data, particularly effective for short sequence lengths. In its original paper, TimeGAN demonstrates its capabilities on samples with sequence lengths of $l = 24$, showcasing limitations of LSTM-based architectures. By contrast, a model like WaveGAN (Donahue et al., 2019), which is built on a convolutional architecture, is trained on significantly longer sequence lengths, with $l = 16384$ at minimum. This contrast highlights the fundamental differences and capabilities between recurrent and convolutional networks.

The limitations of LSTMs in modeling long-term dependencies are not restricted to time series data but also impact their performance in other domains, such as natural language processing (NLP). Early applications of attention mechanisms integrated with recurrent neural networks like LSTMs (Bahdanau, 2014) have largely been replaced by Transformer architectures (Vaswani et al., 2017), which excel in data-rich tasks due to their parallel processing capabilities and expressive attention mechanisms. While Transformer architectures have shown exceptional results in NLP (Radford et al., 2019), their application to time series data remains challenging. This is due in part to the self-attention mechanism's quadratic scaling in memory and computation with sequence length (Katharopoulos et al., 2020), which makes them less practical for very long sequences. Additionally, the inductive bias of Transformers differs from that of recurrent models: Transformers rely on positional encodings to model temporal structure, whereas recurrent architectures such as LSTMs process data sequentially by design, which inherently embeds a sense of temporal order into the model dynamics. This sequential processing makes recurrent models particularly well-suited for long, approximately stationary time series, where preserving temporal continuity over extended horizons can be highly beneficial.

Among the primary approaches for generative modeling of time series, three dominant frameworks have emerged: Generative Adversarial Networks (GANs) (Goodfellow et al., 2020), Variational Autoencoders (VAEs) (Kingma & Welling, 2014; Fabius & Van Amersfoort, 2014), and, more recently, Diffusion Models (Ho et al., 2020). Diffusion Models have demonstrated impressive capabilities in modeling complex data distributions, but their significant computational demands, high latency, and complexity make them less practical for many applications (Yang et al., 2024). Moreover, in terms of practical applications, there are often constraints in both time and computational resources, which limit the feasibility of performing extensive fine-tuning for each individual dataset. A general, well-performing approach that is both simple and efficient is therefore more desirable. In this context, VAEs still stand out for their simplicity and direct approach to probabilistic modeling. In our work, we focus on VAEs and propose a method for training VAEs with recurrent layers to handle longer sequence lengths. We argue that VAEs are particularly suited for generation of time series data, as they explicitly learn the underlying data distribution, making them robust, interpretable, and straightforward to implement.

Our major contributions are:

- We introduce a novel combination of inductive biases, network topology, and training scheme in a recurrent variational autoencoder architecture. Our model integrates approximate time-shift equivariance into a recurrent structure, encouraging invariance to absolute time and thereby providing an inductive bias toward quasi-periodic time series. Unlike existing recurrent or convolutional generative models, our architecture maintains a fixed number of parameters, independent of the sequence

length. We further analyze this behavior through the Echo State Property (ESP), which serves as a diagnostic lens to quantify how strongly the model forgets arbitrary initializations and aligns its dynamics with the input structure.

- We propose a simple yet effective training scheme that subsequently increases the sequence length during training, leveraging the sequence-length-invariant parameterization of our model. We refer to the combination of our architecture with this training scheme as AEQ-RVAE-ST. This scheme mitigates the typical limitations of recurrent layers in capturing long-range dependencies, is particularly suited for quasi-periodic datasets, and contributes significantly to our model's performance.

- We conduct extensive experiments on five benchmark datasets and compare our method against a broad range of strong baselines, including models based on GANs, VAEs, diffusion processes, convolutions, and Transformers. This diverse set covers the most prominent architectural families in time-series generation and ensures a fair and comprehensive evaluation.

- To evaluate generative quality of the long generated sequences, we employ a comprehensive set of evaluation metrics, including Contextualized Fréchet Inception Distance (Context FID), Discriminative Score, and visualizations via PCA and t-SNE.

Our implementation, including preprocessing and model training scripts, is available in branches: main, sine, inductive bias.

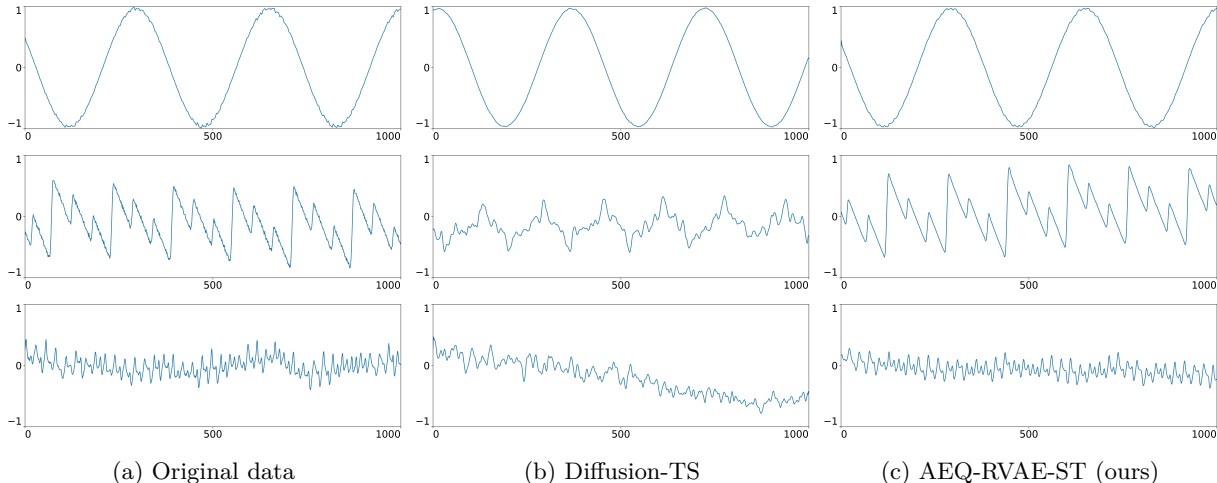

(a) Original data
(b) Diffusion-TS
(c) AEQ-RVAE-ST (ours)

Figure 1: Three excerpts of the electric motor dataset (4.1) with sequence length $l = 1000$: (a) original data, (b) generated by Diffusion-TS (Yuan & Qiao, 2024), (c) generated by our model. Each row corresponds to a different channel. Our model (c) faithfully reproduces key signal characteristics across all channels, including the pronounced voltage peak volatility (row 1), the sawtooth-like DC-bus pattern (row 2), and both the high- and low-frequency components of the stator current (row 3). Diffusion-TS (b) captures the general frequency structure but loses fine-grained details such as the sawtooth shape and high-frequency content.

## 2 Related Work

### 2.1 Deep Generative Models for Time Series

Time-series generation has been explored across various deep generative paradigms, including GANs, VAEs, Transformers, and diffusion models. Early approaches focused on recurrent structures: C-RNN-GAN (Mogren, 2016) used LSTM-based generators and discriminators, while RCGAN (Esteban et al., 2017) introduced label-conditioning for medical time series. TimeGAN (Yoon et al., 2019) combined adversarial training, supervised learning, and a temporal embedding module to better capture temporal dynamics. Around the

same time, WaveGAN (Donahue et al., 2019) introduced a convolutional GAN architecture for raw audio synthesis, illustrating that convolutional models can also be effective for generative tasks in the time domain. TimeVAE (Desai et al., 2021b) further explored this direction by proposing a convolutional variational autoencoder tailored to time-series data. PSA-GAN (Paul et al., 2022) employed progressive growing (Karras et al., 2018), incrementally increasing temporal resolution during training by adding blocks composed of convolution and residual self-attention to both the generator and discriminator. This fundamentally differs from our approach, which extends sequence length rather than resolution.

Recent advances in time-series generation have explored diffusion-based and hybrid Transformer architectures. Diffusion-TS (Yuan & Qiao, 2024) introduces a denoising diffusion probabilistic model (DDPM) tailored for multivariate time series generation. It employs an encoder-decoder Transformer architecture with disentangled temporal representations, incorporating trend and seasonal components through interpretable layers. Unlike traditional DDPMs, Diffusion-TS reconstructs the sample directly at each diffusion step and integrates a Fourier-based loss term. Time-Transformer (Liu et al., 2024) presents a hybrid architecture combining Temporal Convolutional Networks (TCNs) and Transformers in a parallel design to simultaneously capture local and global features. A bidirectional cross-attention mechanism fuses these features within an adversarial autoencoder framework (Makhzani et al., 2016). This design aims to improve the quality of generated time series by effectively modeling complex temporal dependencies.

A common limitation across all these approaches is their focus on relatively short sequence lengths. Many models, including TimeGAN, TimeVAE, and Time-Transformer, are evaluated at $l = 24$. Only the transformer-based Diffusion-TS and PSA-GAN extend this slightly, with ablations up to $l = 256$, leaving the performance on significantly longer sequences largely unexplored.

## 2.2 Recurrent Variational Autoencoders

The Recurrent Variational Autoencoder (RVAE) was introduced by Fabius & Van Amersfoort (2014), combining variational inference with basic RNNs for sequence modeling. In this architecture, the latent space is connected to the decoder via a linear layer, and the sequence is reconstructed by applying a sigmoid activation to each RNN hidden state.[1] We build on this framework by replacing the basic RNNs with LSTMs (or GRUs) and using a repeat-vector mechanism that injects the same latent vector at every time step of the decoder. This design encourages the latent code to encode global sequence properties, while the LSTM handles temporal dependencies. Instead of a sigmoid, we apply a time-distributed linear layer, preserving approximate time-translation equivariance (see Section 3.1).

Unlike dynamic VAEs (dVAE) that use a sequence of latent variables to increase flexibility (Girin et al., 2021), we opt for a single latent vector of fixed size across the entire sequence. This choice reflects our focus on the inductive bias of translational equivariance and stationarity, where the latent code is meant to capture global properties of the sequence while allowing the decoder to model local temporal dynamics. This distinction means that, unlike in dVAE models, the latent code does not change over time, aligning with the assumptions of our model and the goal of preserving global structure while modeling temporal relationships.

# 3 Methods

## 3.1 Stationarity, Time-Shift Equivariance, and ESP

A central challenge in generative modeling of time series is how models handle temporal invariances. Real-world sensor data rarely satisfies strict stationarity. Instead, it often exhibits quasi-periodicity, characterized by similar repeating patterns whose amplitude or frequency may vary slowly over time. Such data can be viewed as approximately stationary over limited horizons, since its statistical properties remain relatively stable under small temporal shifts. This raises the question of time-shift equivariance: whether a model's predictive distribution treats the same local pattern consistently, independent of its absolute position within the sequence.

---

[1] https://github.com/arunesh-mittal/VariationalRecurrentAutoEncoder/blob/master/vrae.py

Recurrent architectures such as LSTMs naturally encourage this behavior through their sequential update mechanism, but in practice true equivariance does not hold, as hidden states may retain information about initial conditions or absolute position. This effect can be studied through the Echo State Property (ESP), which describes the ability of recurrent networks to forget their initialization and converge to a state determined solely by the input sequence.

While ESP is not equivalent to stationarity, it facilitates approximate shift equivariance by removing spurious dependencies on the initial hidden state. After a sufficient washout period, the network state is determined primarily by the recent input sequence rather than by absolute position.

To avoid confusion, we briefly summarize the concepts used in this work:

- **Stationarity (data property):**

  A process $(x_t)$ is strictly stationary if the joint distributions of any two windows $x_{t:t+\ell}$ and $x_{t+\Delta:t+\ell+\Delta}$ are identical for all shifts $\Delta$. In practice, however, most real-world time series are only approximately stationary. A common and practically relevant case is **quasi-periodicity**, where the data exhibit recurring but not perfectly regular patterns, such as oscillations with slowly varying amplitude, phase, or frequency, that give rise to long-term statistical regularities without strict invariance. Following the quasi-periodic anomaly-detection literature (Liu et al., 2022; Zangrando et al., 2022; Tang et al., 2023), we characterize this regime operationally as time series that admit a segmentation into cycles such that consecutive cycles are similar after mild alignment (e.g., small time-warping or phase shift) and normalization, while residual components capture drift, noise, and anomalies. This perspective aligns with how pseudo-periodic streams are handled in the data-stream and segmentation literature (an Tang et al., 2007; Yin et al., 2014).

- **Time-shift equivariance (model property):**

  A model is time-shift equivariant if it treats the same local pattern equivalently, regardless of its absolute position in the sequence. Formally, for strictly stationary data and small shifts $\Delta$, the predictive distributions should satisfy

  $$D(p_\theta(x_{t+1} \mid x_{1:t}), \ p_\theta(x_{t+1+\Delta} \mid x_{\Delta+1:t+\Delta})) \approx 0,$$

  where $D$ denotes a divergence such as Kullback–Leibler or Jensen–Shannon.

- **Echo State Property (ESP, dynamical property of recurrent models):**

  ESP states that the influence of the initial hidden state vanishes over time: for any input sequence $(x_t)$ and any two initializations $(h_0, c_0)$ and $(h'_0, c'_0)$,

  $$\|F_t(x_{1:t}; h_0, c_0) - F_t(x_{1:t}; h'_0, c'_0)\| \ \to \ 0 \quad \text{as } t \to \infty,$$

  where $F_t$ denotes the unrolled recurrence. ESP provides a mechanism for approximate time-shift equivariance, since after a sufficient washout period the hidden state depends only on the input sequence and not on absolute position.

To illustrate this relation more concretely, consider the recurrent transition of an LSTM cell,

$$(h_{i+1}, c_{i+1}) = \hat{f}(x_i, h_i, c_i),$$

which defines a discrete-time dynamical system on the hidden state. Now consider two partially overlapping input sequences $X = [x_0, \ldots, x_n]$ and $X' = [x_1, \ldots, x_n]$, where $X'$ starts one step later but otherwise shares the same continuation. When both sequences are propagated through the recurrence $\hat{f}$, their hidden trajectories initially differ due to the additional update step in $X$. However, under stable dynamics this difference diminishes over time, and

$$\hat{f}(x_k, \hat{f}(x_{k-1}, \ldots, \hat{f}(x_1, \hat{f}(x_0, h, c)))) \ \approx \ \hat{f}(x_k, \hat{f}(x_{k-1}, \ldots, \hat{f}(x_1, h, c))), \tag{1}$$

for sufficiently long sequences. This convergence of hidden trajectories, often referred to as *state forgetting*, is the operational manifestation of the Echo State Property and underlies approximate time-shift equivariance in recurrent models.

## 3.2 Architectural Considerations for Quasi-Periodic Time Series

Given that our focus is on time series data with quasi-periodic behavior, other architectures such as convolutional layers and transformers face specific limitations. Convolutional layers are widely used to build translation-equivariant networks, which makes them highly effective in domains like image processing where pattern recognition should be invariant to position. However, in the context of time series, convolution alone is not inherently designed for sequence generation: upscaling typically increases the resolution of a fixed time window rather than extending the sequence length itself (Paul et al., 2022). This distinction limits the ability of convolutional architectures to generate variable-length time series.

Transformers, on the other hand, excel at capturing long-range dependencies, but their self-attention mechanism scales quadratically with sequence length (Katharopoulos et al., 2020), which makes them computationally demanding for very long series. Moreover, transformers are not inherently translation-equivariant. Instead, they are permutation-equivariant and therefore require explicit positional encodings to represent temporal order. While this flexibility is powerful for text or other symbolic sequences, it contrasts with the requirements of time series generation, where a consistent sense of order and time-shift equivariance are central.

By comparison, recurrent architectures such as LSTMs embed temporal order directly into their model dynamics. They maintain an internal state that evolves sequentially with the data, naturally supporting the kind of approximate time-shift equivariance discussed above.

These properties motivate our choice of recurrent architectures for modeling quasi-periodic time series as studied in prior anomaly-detection work (Liu et al., 2022; Zangrando et al., 2022; Tang et al., 2023). In such data, approximate time-shift equivariance encourages representations that are stable under phase shifts and cycle-to-cycle timing variability, matching the practical need to recognize the same pattern regardless of its temporal position. Our progressive training scheme (Section 3.4) complements this architectural choice by stabilizing learning on long horizons where repetition exists but is not exact due to drift, noise, and anomalies. These conditions are characteristic of quasi-periodic benchmarks and industrial settings described in prior work (Zangrando et al., 2022; Yang et al., 2025).

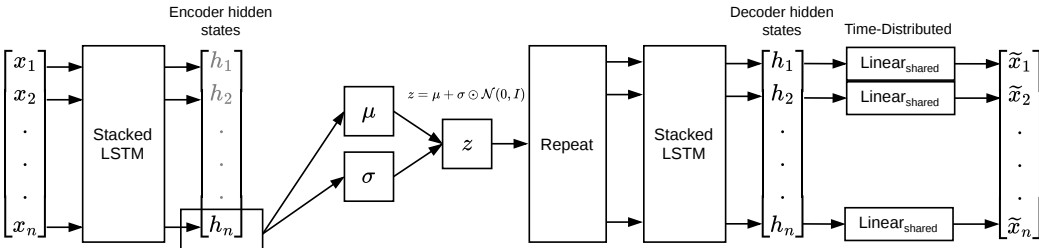

Figure 2: This figure illustrates the architecture of our model. Both the encoder and decoder consist of stacked LSTM layers. The encoder's final hidden state, denoted as $h_n$, is used to compute the parameters $\mu$ and $\log(\sigma^2)$, from which the latent variable $z$ is sampled. The latent variable $z$ is then repeated across all time steps and used as the input to the decoder. The decoder generates the sequence step-by-step, with each time step's hidden state passed through a time-distributed linear layer with shared weights. This is represented by the individual Linear$_{\text{shared}}$ blocks. Throughout the network, approximate equivariance is maintained with respect to time translation, and the number of trainable parameters remains constant regardless of the sequence length. For a detailed layer-by-layer specification of the decoder with the explicit dimensions used in our experiments, see Appendix A.3.

### 3.3 AEQ-RVAE-ST

Our model builds on the Variational Autoencoder (VAE) framework (Kingma & Welling, 2014; Fabius & Van Amersfoort, 2014), which minimises the negative evidence lower bound (ELBO):

$$\mathcal{L}_{\theta,\phi}(x) = \underbrace{-\mathbb{E}_{q_\phi(z|x)}[\log p_\theta(x|z)]}_{\mathcal{L}_E} + \underbrace{D_{\mathrm{KL}}(q_\phi(z|x) \,\|\, p_\theta(z))}_{\mathcal{L}_R}, \tag{2}$$

where $\mathcal{L}_E$ is the reconstruction loss, $\mathcal{L}_R$ the KL-divergence to the prior $p_\theta(z) = \mathcal{N}(0, I)$, and $q_\phi(z|x)$ the approximate posterior (Murphy, 2022).

The inference network $q_\phi(z|x)$ is implemented using stacked LSTM layers. Given the final point in time of a sequence, the output of the last LSTM layer is passed through two linear layers to produce $\mu$ and $\log(\sigma)$. The latent variable is sampled via the reparameterisation trick, $z = \mu + \sigma \odot \epsilon$ with $\epsilon \sim \mathcal{N}(0, I)$. The generative network $p_\theta(x|z)$ then reconstructs the data from $z$. To achieve this, $z$ is repeated across all time steps (using a repeat vector), ensuring that it remains constant and shared throughout the entire sequence:

$$z_t = z \quad \text{for all } t \in \{1, 2, \dots, n\},$$

where $n$ denotes the total number of time steps. The repeat vector is followed by stacked LSTM layers. Finally, a time-distributed linear layer is applied in the output. This layer operates independently at each time step, applying the same linear transformation to the LSTM output at every time step, which can be viewed as a $1 \times 1$ convolution across the time dimension, with shared weights across all time steps.

The time-distributed layer is inherently equivariant with respect to time-translation, preserving temporal structure and shifts over time. Together with our LSTM-based approach and the repeat-vector mechanism, this design ensures that the number of trainable parameters remains independent of the sequence length, while also enabling an adapted training scheme that can accommodate increasing sequence lengths. The architecture is illustrated in Figure 2. Details and hyperparameters are provided in Appendix A.6.

### 3.4 Training scheme for sequence lengths

Building on the principles of time-shift equivariance and state forgetting discussed in Section 3.1, we adopt a progressive training scheme that incrementally increases the sequence length during training. While the recurrent architecture introduced in Section 3.3 provides the necessary structural inductive bias, training on long sequences from the beginning often leads to unstable gradients and poor convergence. Our approach mitigates this by first training on short sequences and gradually extending the sequence length, allowing the model to incrementally adapt to longer temporal dependencies without sacrificing training stability.

Training a recurrent model such as an LSTM to generate consistent long sequences is challenging, as recurrent layers have a limited capacity to preserve information over extended temporal ranges. To facilitate learning over longer horizons and to encourage stable hidden-state dynamics, we employ a progressive training scheme for the AEQ-RVAE-ST model. The dataset is initially divided into short sequences on which the model is first trained, stabilizing optimization and accelerating convergence. After this initial phase, we subsequently increase the sequence length: the dataset is rebuilt into longer chunks, and training continues until the validation loss saturates. This process is repeated iteratively, enabling stable training over increasingly long horizons. Empirically, we find that this scheme improves both convergence stability and final performance compared to training directly on long sequences.

This scheme can be motivated probabilistically. For a time series $x$ of length $l$, hidden features $h$ of length $l$, and a latent vector $z$, we assume a recurrent generative structure:

$$p(x, h, z) = p(z) \prod_{i=1}^{l} p(h_i \mid z, h_{i-1}, \dots, h_1) \, p(x_i \mid h_i).$$

This process can be approximated by restricting dependencies to a finite memory of $t$ steps:

$$p(x, h \mid z) = \prod_{i=1}^{l} p(h_i \mid z, h_{i-1}, \ldots, h_1)\, p(x_i \mid h_i)$$

$$\approx \prod_{i=1}^{l} p(h_i \mid z, h_{i-1}, \ldots, h_{\max(1, i-t)})\, p(x_i \mid h_i). \tag{3}$$

Training on shorter sequences therefore corresponds to learning a truncated approximation of the full generative process. Subsequently extending the sequence length during training relaxes this truncation and allows the model to gradually approximate the full time-shift invariant distribution $p_\theta(x)$. This progressive extension of the training horizon operationalizes the approximate time-shift equivariance discussed earlier, allowing the model to learn stable long-term dynamics in quasi-periodic data.

Based on our experience, we recommend starting at a short sequence length (e.g., $l = 50$ to $l = 100$) and using moderate increments ($\Delta l \leq 150$). Large increments ($\Delta l \geq 300$) tend to degrade performance. In the main experiments, we use increments of $\Delta l = 100$ as a robust default. A detailed sensitivity analysis is provided in Appendix A.1.

## 4 Experiments

In our experiments, we compare the performance of AEQ-RVAE-ST to several baseline models. AEQ-RVAE-ST uses a single, fixed configuration across all datasets and sequence lengths. For the baselines, we adopt official configurations from the respective repositories (e.g., for Diffusion-TS we use the authors' Sine configuration for the Sine dataset); see Appendix A.11.1 for details.

For the training procedure, we started with a sequence length of 100 and subsequently increased it by 100 in each subsequent training phase, until reaching a maximum sequence length of 1000. We compare the performance of the models at sequence lengths of 100, 300, 500, and 1000. To evaluate performance, we employ a combination of short-term consistency measures based on independently generated ELBOs, the discriminative score, and the contextual FID score. Additionally, we perform visual comparisons between the training and generated data distributions using dimensionality reduction techniques such as PCA and t-SNE. All reported results were tested for statistical significance using the Wilcoxon rank-sum test (Wilcoxon, 1992). In cases where the difference was not statistically significant, multiple values are highlighted in bold.

### 4.1 Data Sets

For our experiments we use three multivariate sensor datasets with typical semi-stationary behavior, a synthetic benchmark, and one less quasi-periodic dataset. We specifically selected datasets with a minimum size necessary for training generative models effectively.

**Electric motor (EM)(Wißbrock & Müller, 2025; Mueller, 2024):** This dataset was collected from a three-phase motor operating under constant speed and load conditions. We use only the file H1.5, selected arbitrarily among the available recordings. The data was recorded at a sampling rate of 16 kHz. Out of the twelve initially available channels, four were removed due to discrete behavior or abrupt changes, leaving only smooth, continuous signals. The resulting dataset contains approximately 250,000 datapoints and represents a highly quasi-periodic real-world time series.

**ECG data (ECG)[2] Goldberger et al. (2000):** This dataset contains a two-channel electrocardiogram recording from the MIT-BIH Long-Term ECG Database. It has nearly 10 million time steps of which we use the first 500,000 for training. The signals exhibit clear periodic structure corresponding to cardiac cycles, yet show natural variability in frequency and morphology, including occasional irregularities such as arrhythmias. Our objective is not to produce medically usable data; specialized models are likely more appropriate for medical applications (Neifar et al., 2023).

---

[2]https://physionet.org/content/ltdb/1.0.0/14157.dat

**ETTm2 (ETT)**[3]**:** The ETTm2 dataset contains sensor measurements such as load and oil temperature from electricity transformers, recorded over a two-year period at a coarse sampling rate of four points per hour. While originally proposed for long-term forecasting (Zhou et al., 2021), our analysis suggests that its temporal dynamics are weakly quasi-periodic at best, due to the limited temporal resolution, the short analysis horizon relative to the seasonal cycles, and the small dataset size (69,680 samples).

**Synthetic Sine:** This dataset consists of five independent sine waves with randomly sampled frequencies and initial phases drawn from $\mathcal{N}(0, 0.1)$. The resulting signals are smooth, noise-free, and nearly stationary, serving as a canonical benchmark for generative time-series models (Yoon et al., 2019; Desai et al., 2021b; Yuan & Qiao, 2024).

**MetroPT3 (Davari et al. (2021)):** The MetroPT3 dataset consists of multivariate time-series data from analogue and digital sensors installed on a compressor, originally collected for predictive maintenance and anomaly detection. The data were logged at $1\,\mathrm{Hz}$. Similar to the Electric Motor dataset, we removed non-continuous or discrete signals. Out of the original 1.5 million time steps, we used the first 500,000. Recurring patterns in this dataset are frequently interrupted by phases of flat signals, leading to irregular temporal dynamics.

## 4.2 Baseline Models

We compare against five established generative models spanning GANs, VAEs, diffusion models, and Transformer-based architectures: TimeGAN (Yoon et al., 2019), a recurrent GAN; WaveGAN (Donahue et al., 2019), a convolutional GAN originally designed for raw audio; TimeVAE (Desai et al., 2021b), a convolutional VAE; Diffusion-TS (Yuan & Qiao, 2024), a diffusion model with Transformer-based trend-seasonal decomposition; and Time-Transformer (Liu et al., 2024), an adversarial autoencoder combining TCNs and Transformers. None of these baselines enforce approximate time-shift equivariance by design. A detailed description of each model's architecture and equivariance properties is provided in Appendix A.10.

## 4.3 Emphasizing Inductive Bias with Echo State Property (ESP)

The ESP provides a useful lens to analyze inductive bias in recurrent generative models. Formally, ESP states that when driven by the same input sequence, hidden states forget arbitrary initial conditions and converge to a unique input-determined trajectory (Jaeger, 2001; Manjunath & Jaeger, 2013). Note that standard LSTMs do not guarantee ESP in general. Our measurements are empirical indicators of contraction rather than a formal guarantee (Yildiz et al., 2012; Buehner & Young, 2006).

Interpreting ESP in our setting leads to three important insights:

**ESP as forgetting irrelevant information:** Strong ESP does not mean that the network indiscriminately forgets all information, but specifically that it suppresses dependence on arbitrary initializations. Once washout has occurred, the hidden states become determined primarily by the input. This aligns well with nearly stationary or quasi-periodic data, where invariance to absolute time is desirable.

**ESP versus meaningful long-term memory:** A model without ESP may appear to "retain" information longer, but what is retained is often dependence on the random initialization rather than useful structure in the input sequence. Conversely, moderate ESP allows the model to forget initialization artifacts while still preserving long-term dependencies driven by the input. Thus, ESP should not be interpreted as the opposite of memory capacity, but rather as the ability to separate meaningful input-driven memory from spurious initialization effects.

**Inductive bias for stationarity:** In generative modeling of time series, ESP encourages the network to emphasize relative temporal patterns over absolute time indices. This induces an inductive bias toward stationarity-like behavior: repeated patterns are treated consistently regardless of where they occur in the sequence. At the same time, the property is only approximate in our trained models, allowing flexibility to retain non-stationary structure (e.g., trends or irregular variations) when present in the data.

---

[3]https://github.com/zhouhaoyi/ETDataset

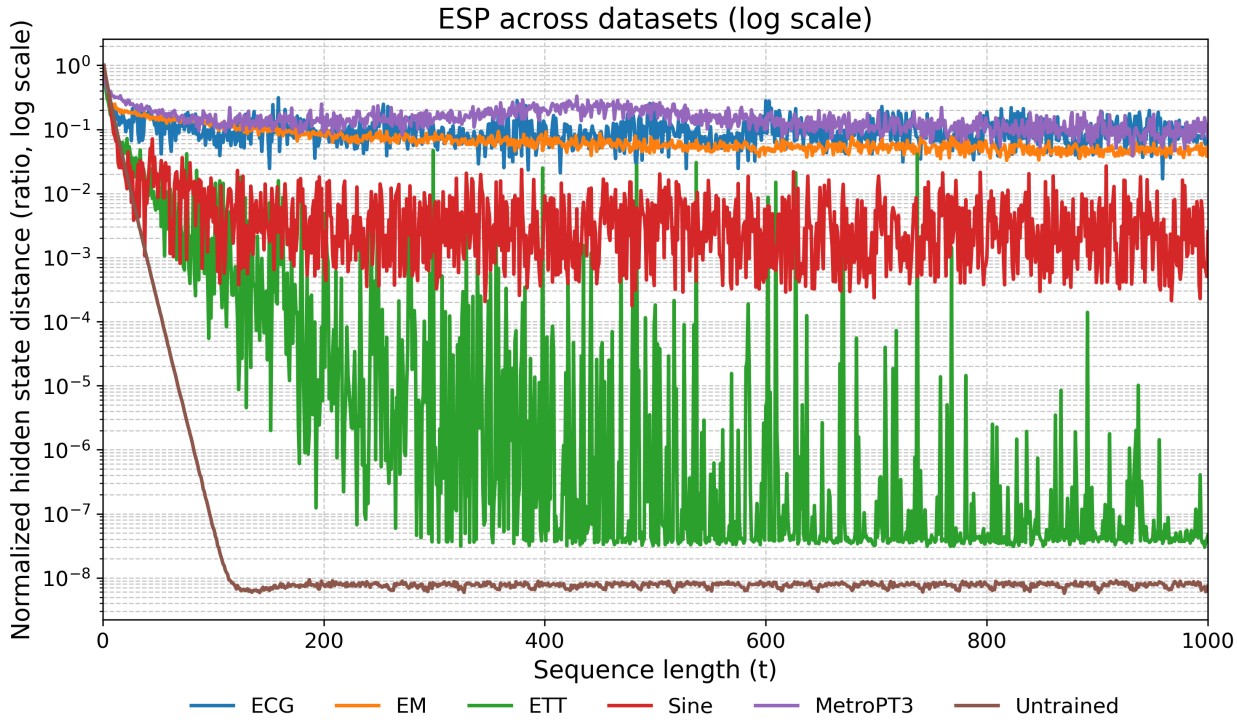

Figure 3: Echo State Property (ESP) analysis across datasets (log scale). The x-axis shows sequence length $t$ and the y-axis the normalized hidden-state distance $r(t) = d(t)/d(0)$, where $d(t) = \mathbb{E}_{z, (h_0^\top), (h_0^{\top\prime})}[\| h_t^\top - h_t^{\top\prime} \|_2]$. Here $h_t^\top$ denotes the hidden state of the top LSTM layer only (no cell states are used). Random initializations are applied only to the top layer; all lower layers use identical fixed initializations across runs. By construction we set $r(0) = 1$ (the curve is normalized to the initial distance). The expectation is approximated by sampling 10 latent vectors $z$; for each $z$ one input sequence is generated and $d(t)$ is averaged over 20 independent pairs of random top-layer initial states $(h_0^\top, h_0^{\top\prime})$. On the logarithmic scale, exponential contraction appears as straight lines. The untrained model indeed shows such monotonic exponential decay, with $r(t)$ converging into the range of $10^{-9}$ and remaining stable thereafter, reflecting trivial washout due to random initialization. In contrast, trained models display weaker contraction with residual variability in the curves. ETTm2 reaches about $10^{-7}$, the synthetic sine dataset around $10^{-3}$, while ECG, EM, and MetroPT3 remain higher. This illustrates how training balances ESP with the preservation of long-term temporal structure. All hidden states are calculated based on the weights of the models with $l = 1000$.

As shown in Fig. 3, we observe contraction across all datasets, though with varying strength. Trained models exhibit weaker contraction than the untrained baseline, with ETTm2 showing the strongest and ECG, EM, and MetroPT3 the weakest contraction among trained models. This illustrates how training counterbalances the architectural bias by preserving input-driven dependencies where useful, rather than enforcing unconditional washout.

The particularly strong contraction observed on ETTm2 is likely due to the combination of a coarse temporal resolution of four samples per hour, an analyzed horizon of 1000 steps (approximately 10 days), and a limited total span of about two years. Together, these factors make it difficult to learn meaningful long-term seasonal dependencies, so that the model instead forgets initial states rapidly, producing the appearance of strong ESP.

### 4.4 Evaluation by Context-FID Score

To evaluate the distributional similarity between real and generated time series, we use the Context-FID score (Paul et al., 2022), a variant of the Fréchet Inception Distance (FID) commonly used in image generation. In this adaptation, the original Inception network is replaced by TS2Vec (Yue et al., 2022), a self-supervised representation learning method for time series. The score is computed by encoding both real and generated sequences with a pretrained TS2Vec model and calculating the Fréchet distance between the resulting feature distributions. Lower scores indicate that the synthetic data better matches the distribution of the real data.

Table 1 reports the Context-FID scores across different sequence lengths and datasets.

Across the different sequence lengths, AEQ-RVAE-ST consistently outperforms all comparison models on the Electric Motor, ECG, and especially the Sine datasets starting from $l = 300$. These datasets exhibit high quasi-periodicity, which aligns well with the inductive biases of our approach. On the lesser quasi-periodic datasets MetroPT3 and ETT, our model remains competitive, with TimeVAE surpassing it at $l = 1000$ for both datasets. Additionally, for MetroPT3, Diffusion-TS outperforms our model at $l = 500$.

### 4.5 Evaluations by Discriminative Score

The discriminative score $\mathcal{D}$ was introduced by (Yoon et al., 2019) as a metric for quality evaluation of synthetic time series data. For the discriminative score a simple 2-layer RNN for binary classification is trained to distinguish between original and synthetic data. Implementation details are in the appendix A.12. It is defined as $\mathcal{D} = |0.5 - a|$, where $a$ represents the classification accuracy between the original test dataset and the synthetic test dataset that were not used during training. The best possible score of 0 means that the classification network cannot distinguish original from synthetic data, whereas the worst score of 0.5 means that the network can easily do so.

The discriminative score provides particularly meaningful insights when it allows for clear distinctions between models, which is best achieved by avoiding scenarios where the score consistently reaches its best or worst possible values across different models. To ensure consistency, we used the same fixed number of samples for training the discriminator across all experiments, regardless of sequence length. This fixed sample size was found to be suitable for our experimental setup.

As shown in Table 2, the Discriminative Score yields a less clear-cut picture compared to other evaluation metrics. The Wilcoxon rank-sum test reveals that in several cases, performance differences between models are not statistically significant.

On the Electric Motor dataset, AEQ-RVAE-ST achieves the best performance from $l = 300$ onwards. For the ECG dataset, AEQ-RVAE-ST outperforms all other models at $l = 1000$, while for shorter sequence lengths, its performance is comparable to that of Diffusion-TS. On the ETT dataset, AEQ-RVAE-ST, TimeVAE, and Diffusion-TS perform similarly well across all sequence lengths, with no statistically significant differences. The Sine dataset exhibits more nuanced behavior: Diffusion-TS performs best at $l = 100$; at $l = 300$, AEQ-RVAE-ST, TimeVAE, and Diffusion-TS perform comparably; and from $l = 500$ onwards, AEQ-RVAE-ST achieves the best results. For the MetroPT3 dataset, AEQ-RVAE-ST is best at $l = 100$, while Diffusion-TS slightly outperforms all other models at longer sequence lengths.

### 4.6 Evaluation by PCA

In this section, we evaluate the quality of the generated time series using PCA (Hotelling, 1933). The idea is to train PCA on the original data, project it into a lower-dimensional space, and apply the same transformation to the synthetic data to assess distributional alignment. While widely used for identifying structural similarities, this technique does not account for temporal dependencies within the sequences. Additional t-SNE (Hinton & Van Der Maaten, 2008) visualizations are provided in Appendix A.14.

These common techniques complement earlier methods that primarily assessed the sample quality of the models. For brevity, we present the results of four selected experiments in the main paper, as all experiments consistently yield the same findings. These four experiments include PCA plots on the EM dataset and on the

Table 1: **FID score** of synthetic time series for six models (see 4.2), computed on the five datasets (see 4.1) at sequence lengths $l = 100$, $l = 300$, $l = 500$, and $l = 1000$. Lower scores indicate better performance. Each score is based on 5000 generated samples, each evaluated (trained) 15 times, and reported with 1-sigma confidence intervals. AEQ-RVAE-ST consistently outperforms all baselines on the highly periodical Electric Motor, ECG, and Sine datasets starting from $l = 300$. On the less periodical MetroPT3 and ETT datasets, performance is more competitive, with TimeVAE and Diffusion-TS outperforming our model at certain sequence lengths.

| Dataset | Model | Sequence lengths | | | |
| | | 100 | 300 | 500 | 1000 |
| --- | --- | --- | --- | --- | --- |
| Electric Motor | **AEQ-RVAE-ST (ours)** | 0.35±0.04 | **0.12±0.01** | **0.10±0.01** | **0.24±0.02** |
| | TimeGAN | 1.03±0.07 | 3.77±0.30 | 3.07±0.24 | 33.7±1.69 |
| | WaveGAN | 0.55±0.04 | 0.75±0.07 | 0.87±0.14 | 1.41±0.24 |
| | TimeVAE | 0.16±0.01 | 0.97±0.11 | 1.06±0.14 | 1.19±0.09 |
| | Diffusion-TS | **0.04±0.00** | 0.69±0.06 | 1.10±0.11 | 1.93±0.13 |
| | Time-Transformer | 2.19±0.16 | 45.4±1.57 | 44.5±2.67 | 65.7±2.86 |
| ECG | **AEQ-RVAE-ST (ours)** | **0.08±0.01** | **0.09±0.02** | **0.14±0.02** | **0.46±0.06** |
| | TimeGAN | 26.8±6.89 | 48.0±6.26 | 47.2±5.91 | 34.0±3.43 |
| | WaveGAN | 1.54±0.19 | 1.56±0.14 | 1.54±0.13 | 1.51±0.16 |
| | TimeVAE | 0.26±0.02 | 0.89±0.07 | 1.07±0.10 | 1.30±0.08 |
| | Diffusion-TS | 0.16±0.01 | 0.28±0.03 | 0.52±0.03 | 3.74±0.22 |
| | Time-Transformer | 1.34±0.11 | 29.7±1.78 | 33.0±2.28 | 40.3±2.44 |
| ETT | **AEQ-RVAE-ST (ours)** | **0.58±0.05** | **0.65±0.07** | **0.79±0.07** | 1.82±0.16 |
| | TimeGAN | 1.51±0.19 | 5.76±0.43 | 13.7±1.28 | 17.7±1.57 |
| | WaveGAN | 3.49±0.22 | 3.90±0.37 | 4.38±0.39 | 4.94±0.42 |
| | TimeVAE | 0.66±0.08 | 0.72±0.08 | 0.97±0.10 | **1.56±0.14** |
| | Diffusion-TS | 0.90±0.11 | 1.18±0.18 | 2.16±0.17 | 2.55±0.27 |
| | Time-Transformer | 1.28±0.14 | 20.1±1.22 | 22.1±1.96 | 47.9±5.28 |
| Sine | **AEQ-RVAE-ST (ours)** | 0.33±0.04 | **0.34±0.02** | **0.46±0.03** | **0.42±0.03** |
| | TimeGAN | 7.70±0.32 | 6.01±0.34 | 7.96±0.37 | 21.8±1.25 |
| | WaveGAN | 1.87±0.10 | 2.09±0.13 | 2.81±0.22 | 3.36±0.27 |
| | TimeVAE | 0.24±0.02 | 0.55±0.05 | 1.26±0.14 | 3.03±1.00 |
| | Diffusion-TS | **0.06±0.00** | 1.52±0.13 | 0.74±0.04 | 2.66±0.20 |
| | Time-Transformer | 0.31±0.02 | 4.10±0.21 | 51.2±1.94 | 74.5±3.85 |
| MetroPT3 | **AEQ-RVAE-ST (ours)** | **0.26±0.04** | **0.65±0.07** | 2.81±0.37 | 2.84±0.22 |
| | TimeGAN | 5.79±0.32 | 10.1±0.79 | 18.6±1.06 | 35.1±3.74 |
| | WaveGAN | 1.14±0.09 | 1.82±0.12 | 2.04±0.16 | 2.43±0.18 |
| | TimeVAE | 0.67±0.05 | 1.32±0.13 | 2.02±0.29 | **2.08±0.31** |
| | Diffusion-TS | 1.07±0.06 | 1.17±0.12 | **1.82±0.09** | 6.97±0.75 |
| | Time-Transformer | 2.28±0.24 | 5.25±0.46 | 22.9±1.45 | 352±66.1 |

ECG dataset, each with sequence lengths of $l = 100$ and $l = 1000$ (see figure 4). The full set of experiments is provided in Appendix A.14.

The visual inspection of the PCA plots for the EM dataset with a sequence length of $l = 100$ reveals no significant differences in the distributions of the models, with Time-Transformer showing a slightly less pronounced overlap compared to the other models. However, as the sequence length increases to l = 1000, the performance differences between the models become clearly visible. Interestingly, the PCA at this length exhibits a circular pattern, indicating the periodic characteristics of the dataset. Among the models,

Table 2: **Discriminative score** of synthetic time series for six models (see 4.2), computed on the five datasets (see 4.1) at sequence lengths $l = 100$, $l = 300$, $l = 500$, and $l = 1000$. A lower score indicates better performance. Each score is based on 15 independent discriminator runs and reported with 1-sigma confidence intervals. AEQ-RVAE-ST performs best on the Electric Motor dataset from $l = 300$ onward and significantly outperforms all models on ECG at $l = 1000$, while showing comparable performance to Diffusion-TS at shorter lengths. For the ETT and Sine datasets, multiple models perform similarly depending on the sequence length. On MetroPT3, AEQ-RVAE-ST is best at $l = 100$, while Diffusion-TS dominates for longer sequences. In cases without statistically significant differences (Wilcoxon rank-sum test), multiple scores are highlighted in bold.

| Dataset | Model | Sequence lengths | | | |
| | | 100 | 300 | 500 | 1000 |
|---|---|---|---|---|---|
| Electric Motor (EM) | **AEQ-RVAE-ST (ours)** | **.121±.021** | **.032±.018** | **.038±.018** | **.085±.015** |
| | TimeGAN | .338±.030 | .477±.018 | .486±.013 | .500±.000 |
| | WaveGAN | .352±.009 | .416±.009 | .425±.011 | .444±.011 |
| | TimeVAE | .268±.214 | .226±.176 | .185±.083 | .152±.047 |
| | Diffusion-TS | **.112±.056** | .327±.130 | .396±.085 | .434±.084 |
| | Time-Transformer | .334±.098 | .500±.000 | .500±.000 | .500±.000 |
| ECG | **AEQ-RVAE-ST (ours)** | **.012±.011** | **.009±.008** | **.016±.014** | **.009±.010** |
| | TimeGAN | .466±.125 | .500±000 | .500±.000 | .500±000 |
| | WaveGAN | .306±.155 | .300±.201 | .402±.153 | .298±.217 |
| | TimeVAE | **.034±.066** | .058±.120 | .131±.181 | .153±.177 |
| | Diffusion-TS | **.007±.007** | **.016±.016** | **.010±.015** | .382±.145 |
| | Time-Transformer | .216±.107 | .500±.000 | .496±.014 | .499±.002 |
| ETT | AEQ-RVAE-ST (ours) | .179±.034 | **.172±.105** | **.189±.049** | **.132±.147** |
| | TimeGAN | **.107±.075** | **.160±.113** | .270±.106 | .320±.120 |
| | WaveGAN | .362±.080 | .345±.113 | .377±.099 | .385±.060 |
| | **TimeVAE** | **.118±.110** | **.140±.053** | **.167±.040** | **.068±.051** |
| | Diffusion-TS | .204±.086 | **.173±.063** | **.151±.055** | **.122±.051** |
| | Time-Transformer | .198±.169 | **.179±.116** | .408±.137 | .500±.000 |
| Sine | **AEQ-RVAE-ST (ours)** | .069±.015 | **.113±.059** | **.080±.044** | **.021±.013** |
| | TimeGAN | .465±.130 | .457±.050 | .491±.005 | .497±.005 |
| | WaveGAN | .187±.036 | .367±.073 | .449±.025 | .449±.034 |
| | TimeVAE | .161±.092 | **.160±.124** | .272±.129 | .347±.144 |
| | Diffusion-TS | **.035±.014** | **.182±.163** | .294±.109 | .428±.105 |
| | Time-Transformer | .173±.019 | .491±.004 | .499±.001 | .500±.000 |
| MetroPT3 | AEQ-RVAE-ST | **.098±.066** | .367±.109 | .423±.074 | .496±.004 |
| | TimeGAN | .428±.041 | .498±.002 | .499±.001 | .499±.001 |
| | WaveGAN | .432±.042 | .494±.005 | .497±.002 | .497±.003 |
| | TimeVAE | .279±.103 | .438±.070 | .488±.024 | .495±.004 |
| | **Diffusion-TS** | .139±.025 | **.251±.022** | **.319±.015** | **.486±.012** |
| | Time-Transformer | .473±.007 | .493±.005 | .500±.000 | .500±.000 |

AEQ-RVAE-ST demonstrates the highest degree of overlap between the original and synthetic data, fitting the circular pattern without outliers. Diffusion-TS performs almost equally well, with slightly less overlap compared to AEQ-RVAE-ST (see figure 1 for visual comparison of the models). WaveGAN shows only a few outliers near the circular pattern. TimeVAE synthetic points further fill the circle, leading to greater deviation from the original data distribution. The PCA plots for the ECG dataset provide a detailed view of models' performances. At $l = 100$, AEQ-RVAE-ST, TimeVAE, and Diffusion-TS perform equally well, showing a

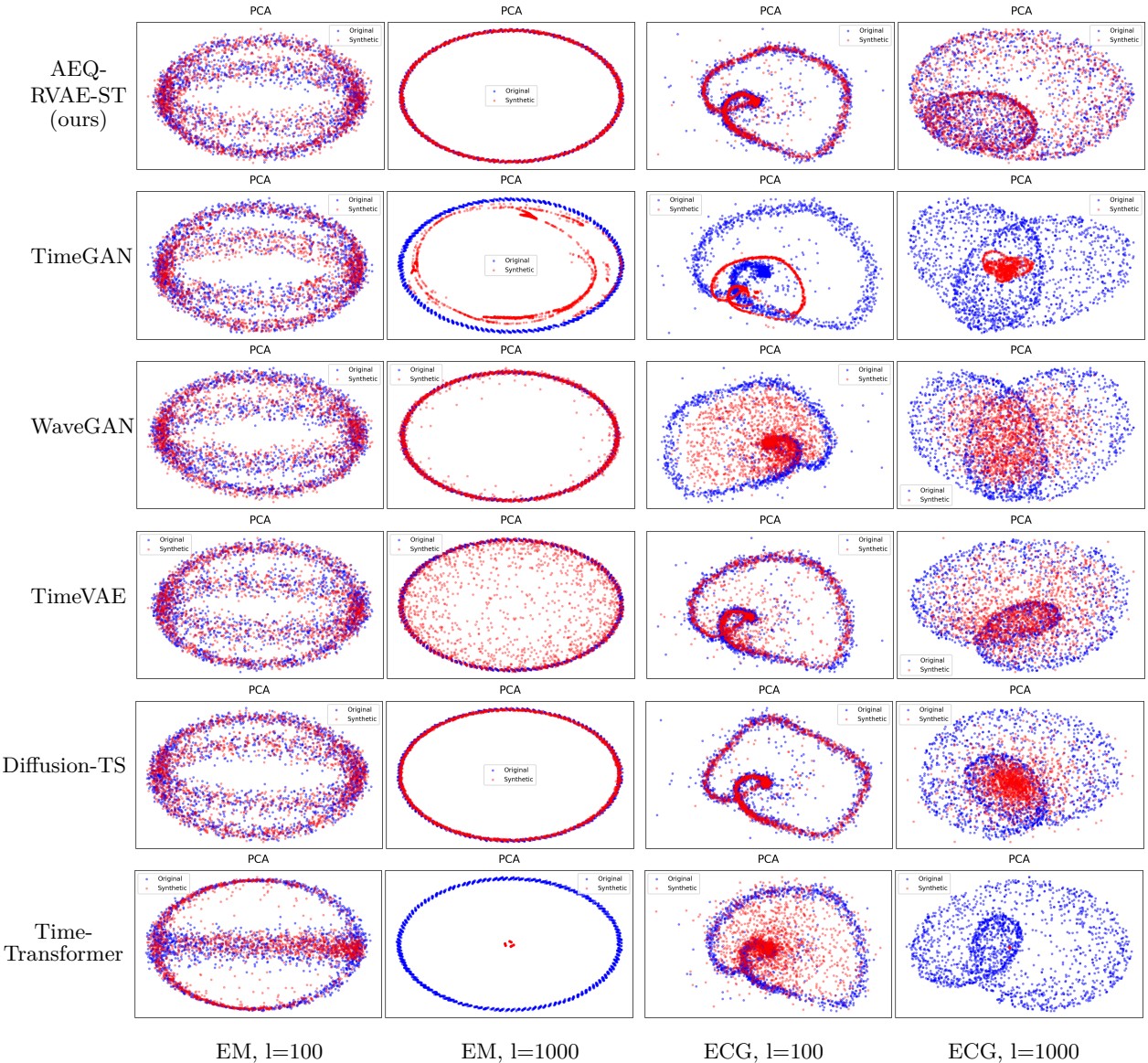

Figure 4: PCA plots for the EM and ECG datasets at sequence lengths of $l = 100$ and $l = 1000$. The higher the overlap between original and synthetic points, the better. For the EM dataset at $l = 100$, no significant differences are observed, with Time-Transformer showing slightly less overlap. At $l = 1000$, the circular pattern reflects the periodic characteristics of the dataset, with AEQ-RVAE-ST demonstrating the best overlap, closely followed by Diffusion-TS. WaveGAN and TimeVAE show some deviations, while TimeGAN exhibits almost no overlap. For the ECG dataset at $l = 100$, AEQ-RVAE-ST, TimeVAE, and Diffusion-TS show strong overlap, while WaveGAN and Time-Transformer exhibit less, and TimeGAN shows almost none. At $l = 1000$, AEQ-RVAE-ST performs best, followed by WaveGAN and TimeVAE, with Diffusion-TS performing worse and TimeGAN and Time-Transformer showing minimal variability and no significant overlap.

strong overlap with the original data. WaveGAN and Time-Transformer show less overlap, and TimeGAN demonstrates almost no overlap at all. At $l = 1000$, AEQ-RVAE-ST achieves the best performance, with the original data being very well represented. This is followed by WaveGAN and TimeVAE, where the synthetic data points cluster together, but with less coverage of the original distribution. Diffusion-TS performs

noticeably worse, while TimeGAN and Time-Transformer show almost no overlap, with the generated data exhibiting minimal variability.

### 4.7 Training scheme ablations

In this experiment, we compare the effectiveness of our proposed training approach against the conventional training method on the same network topology. Our comparison metric is the Evidence Lower Bound (ELBO), calculated for the original dataset $X \in \mathbb{R}^{n_s \times l \times c}$ where $n_s$ represents the numbers of samples, $l$ denotes the sequence length, and $c$ the number of channels. We calculate it as

$$\mathcal{E}(X) = \frac{1}{n_s} \sum_{i=0}^{n_s-1} \text{ELBO}_{\text{norm}}\left(\mathcal{L}_{\theta,\phi}(X_i)\right), \tag{4}$$

where $\mathcal{L}_{\theta,\phi}$ is the loss of the trained model itself. Simply speaking, it is the typical model evaluation on a dataset, but converted to $\text{ELBO}_{\text{norm}}$ (see Appendix A.7). We run this comparison on all datasets with a sequence length of 1000, which is particularly long and challenging. It is the maximum sequence length used in any of the previous experiments. For each of the following training schemes, we do 10 repetitions:

(i) **Conventional train**: One trains the model for a predefined sequence length of $l = 1000$

(ii) **Subsequent train**: The training procedure begins with a sequence length of $l = 100$ and continues until the stopping criteria are met. Afterward, we increase the sequence length by 100 and retrain the model, repeating this process until we complete training with a sequence length of $l = 1000$.

Table 3: Comparison of the effectiveness of our proposed training approach versus the conventional method. The performance metric is the $\text{ELBO}_{\text{norm}}$ as described in Appendix A.7. On each dataset and model we repeated the experiments $n = 10$ times. The 1-sigma confidence intervals describe the results between the independently trained models.

| Train method | EM | ECG | ETTm2 | Sine | MetroPT3 |
|---|---|---|---|---|---|
| conventional train | 0.094±0.004 | 0.103±0.000 | 0.174±0.016 | -0.837±0.566 | -0.140±0.061 |
| subsequent train | **0.218±0.004** | **0.201±0.004** | **0.217±0.012** | **0.194±0.010** | **0.142±0.019** |

As shown in Table 3, the subsequent training scheme (ii) consistently outperforms the conventional training scheme (i) across all datasets, with statistically significant improvements ($p < 0.002$). The largest performance gain is observed on the Sine dataset, where the model's ability to capture sinusoidal patterns improves substantially. In Figure 5, representative samples for each model are shown for a sequence length of $l = 1000$ on the sine dataset. AEQ-RVAE-ST is the only model that can generate proper and consistent sine curves, which are characteristic of the dataset. The Sine dataset, as a clear example of a periodic and almost stationary time series, supports our hypothesis that the AEQ-RVAE-ST model benefits from an inductive bias towards periodicity that enables the model effectively generate consistent, high-quality long-range sequences in such scenarios. Further ablation studies on the sensitivity of the subsequent training scheme to different sequence-length increments, as well as a more detailed analysis of training schedules, are provided in Appendix A.1.

## 5 Discussion

In this paper, we present a hypothesis-driven examination of modeling long time series using approximately time-shift-equivariant architectures. Our central hypothesis is that quasi-periodic time series benefit from an inductive bias that promotes temporal consistency and invariance to absolute time. Approximate time-shift equivariance enables a model to recognize and reproduce recurring temporal patterns across different positions in a sequence, which is particularly important for data with oscillatory or repeating structures.

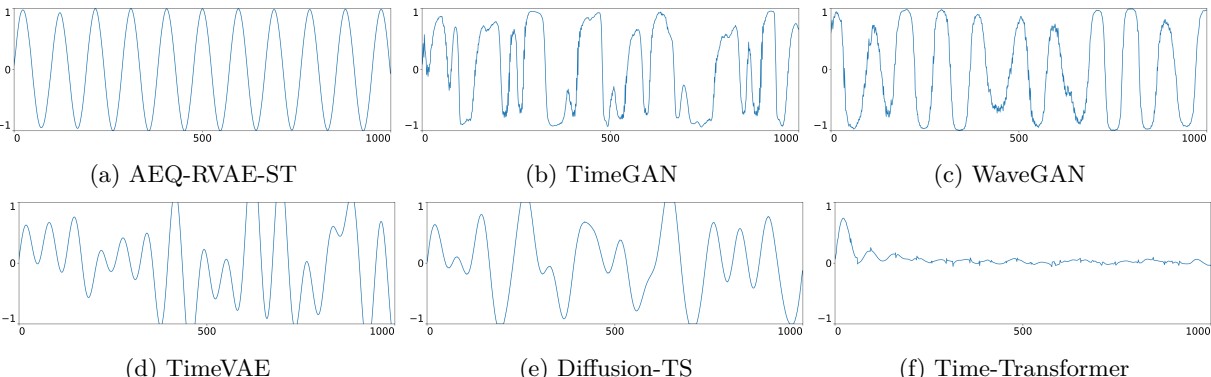

Figure 5: Representative samples for each model at a sequence length of $l = 1000$ on the sine dataset . AEQ-RVAE-ST is the only model capable of consistently generating accurate sinusoidal trajectories, demonstrating its ability to capture the strictly periodic characteristics of the data. For additional samples generated by our model on this dataset, we refer the reader to Figure 9 in the appendix.

While the recurrent layers in our model provide only partial shift equivariance, the overall architecture maintains a consistent transformation behavior across time, leading to two main advantages: (1) an inductive bias that aligns with the characteristics of quasi-periodic and slowly varying temporal dynamics, and (2) a parameterization independent of sequence length, which allows the model to scale efficiently to longer time horizons.

These properties allow the model to exploit temporal regularities more effectively during training and support our interpretation that approximately equivariant recurrent architectures provide a suitable inductive bias for modeling quasi-periodic time series.

In our experiments, we compared AEQ-RVAE-ST with several state-of-the-art generative models across five benchmark datasets. Three of these (Electric Motor, ECG, and Sine) exhibit strong quasi-periodicity, while ETT and MetroPT3 show greater temporal variability, though still containing recurring signal components typical of sensor-based data. On the quasi-periodic datasets, our model consistently outperformed all baselines, especially as sequence length increased, as reflected by the Context-FID and Discriminative Score. On the more irregular datasets, it remained competitive across most configurations. Latent-space visualizations using PCA and t-SNE further confirmed that our model captures the global structure of the data more faithfully than the baselines.

In Section A.2, we demonstrate that a model trained on sequences of length $l = 1000$ can generate coherent samples of arbitrary length, illustrated for $l = 5000$. Together with the results on the Echo State Property (ESP) and state forgetting, these findings lend further support to our theoretical assumption (see Equation 1) that, for sufficiently long sequences, the hidden and cell states converge toward trajectories determined by the input dynamics rather than by initial conditions.

Our findings confirm the effectiveness of the proposed approach and open several promising directions for future research. The methodology could be extended to other model classes, such as diffusion-based generative architectures.

## 5.1 Limitations

Our approach is designed for quasi-periodic time series as characterized in Section 3.1: data that exhibit recurring but imperfectly regular patterns, such as oscillations with slowly varying amplitude, phase, or frequency. While this inductive bias is advantageous for such data, it entails several limitations.

Time series with persistent trends, regime changes, or structural breaks violate the approximate stationarity assumption underlying our approach. Similarly, sparse event-driven time series (e.g., point processes, transaction data) do not exhibit the recurring motifs that our inductive bias exploits. Our model also requires

sufficient training data to learn stable long-horizon dynamics. The ETTm2 dataset used in our experiments (69,681 time steps) represents a borderline case, especially given that seasonal effects in this data unfold over longer time scales than the available data can capture.

Additionally, our model performs best on smoothly varying, high-resolution signals. Low temporal resolution can limit the ability to capture fine-grained temporal structure (e.g., ETTm2 with 4 samples per hour). Time series with abrupt transitions or discontinuities also deviate from our assumptions. MetroPT3 illustrates this challenge: several channels exhibit frequent sharp drops (see Figure 10, TP2, H1, Motor_current), which can dominate the dynamics and reduce the benefit of our inductive bias.

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

# A  Appendix

## A.1  Training Scheme Ablations

Table 4: Sensitivity of subsequent training to different sequence-length increments on the Electric Motor and Sine datasets. Performance is measured via $\text{ELBO}_{\text{norm}}$ (Appendix A.7). Each entry reports the mean over $n = 5$ independently trained models, with 1-sigma confidence intervals. Higher is better.

| Subsequent schedule (sequence length $l$) | EM | Sine |
|---|---|---|
| $50, 100, 150, \ldots, 1000$ | **0.221±0.004** | 0.200±0.005 |
| $100, 200, 300, \ldots, 1000$ | 0.218±0.004 | 0.194±0.010 |
| $100, 250, 400, 550, \ldots, 1000$ | **0.221±0.005** | **0.202±0.004** |
| $100, 400, 700, 1000$ | 0.216±0.003 | 0.156±0.024 |
| $100, 550, 1000$ | 0.216±0.002 | 0.141±0.009 |
| $1000$ (no subsequent train) | 0.094±0.004 | -0.837±0.566 |

For a fair comparison across schedules, we use a same warm-start protocol in all experiments: we always start training at a short sequence length (typically $l = 100$; for the +50 schedule we start at $l = 50$) before applying larger sequence length increments. In our experience, starting directly with large sequence lengths (or making large jumps without this warm start) leads to substantially worse optimization and less stable training.

For each training schedule in Table 4, we train five independently initialized models. Table 4 indicates that performance is relatively robust for small-to-moderate increments, while large jumps degrade $\text{ELBO}_{\text{norm}}$, most notably on Sine. In particular, once the step size becomes large (roughly $\Delta l \gtrsim 300$), performance degrades markedly. Moreover, on both Electric Motor and Sine, the schedules with increments of 50 and 150 outperform the 100-step schedule. Overall, the 150-step schedule yields the best performance across the two datasets considered.

Finally, Table 4 also includes the baseline that trains directly at $l = 1000$ without subsequent training. This baseline performs substantially worse on both datasets, highlighting that the subsequent train is critical for successful learning at long horizons.

## A.2  Extended Time Series

In this section, we provide qualitative examples of generated time series by our model for each of the five datasets used in our evaluation: Electric Motor, ECG, ETT, Sine, and MetroPT3. All samples were

generated with a fixed sequence length of $l = 5000$, using model weights trained on sequences up to $l = 1000$. This allows us to assess the model's ability to generalize and synthesize plausible data beyond the training horizon.

The results illustrate how well the model maintains the structure of the original data when generating extended sequences:

- For time series with higher quasi-periodicity (Electric Motor, ECG, and Sine), the key patterns continue to be synthesized plausibly beyond the training length. In these cases, a stable state emerges, characterized by repeating, but not identical, patterns (see Figures 6, 7, and 9).

- In the Sine dataset, sinusoidal curves are extended effectively, with only a slight reduction in amplitude observable in some channels. (Figure 9).

- For the less quasi-periodic time series (ETT and MetroPT3), a clear degradation in synthesis quality is observed beyond the trained length. In both cases, the model produces repetitive, flatline-like patterns with low variation, and characteristic structures are no longer preserved (Figures 8 and 10).

These qualitative results support the quantitative findings and further highlight the model's ability to generalize well on quasi-periodical data, while revealing its limitations on more dynamic datasets.

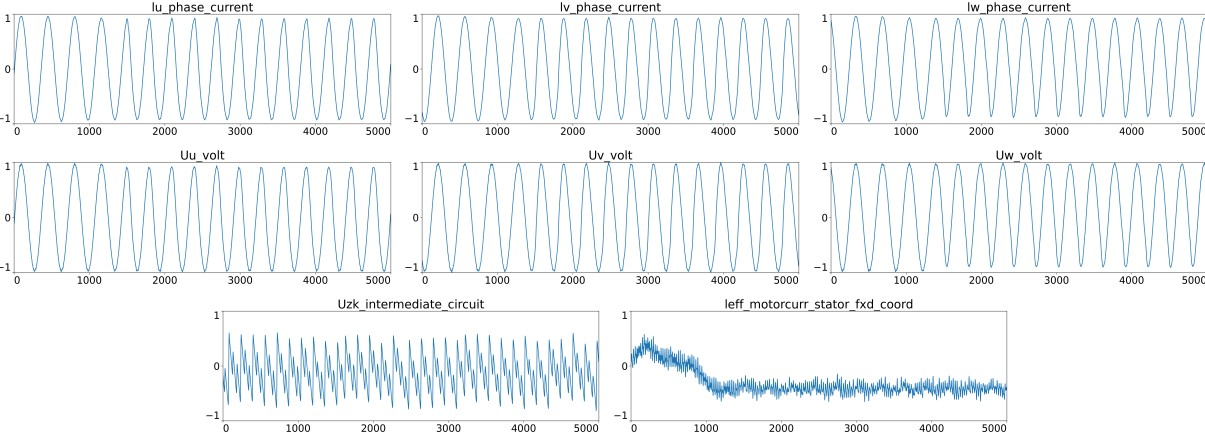

Figure 6: Example of a generated time series sample of length $l = 5000$ from the Electric Motor dataset. The model was trained on sequences up to $l = 1000$. The main characteristics of the dataset continue to be well synthesized in the extended sample. During generation, the model reaches a stable state in which the output patterns kind of repeat. As a result, slower trends, especially visible in the *leff motorcurr stator fxd coord* channel, are not fully reflected in the synthesis.

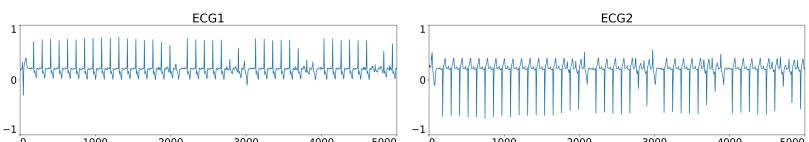

Figure 7: Example of a generated time series sample of length $l = 5000$ from the two-channel ECG dataset. The model was trained on sequences up to $l = 1000$. The key characteristics of the data, particularly the heartbeat-like patterns across both channels, continue to be well synthesized in the extended sequence. Still, a stable state emerges, with periodic patterns that, while not identical, remain strongly similar over time.

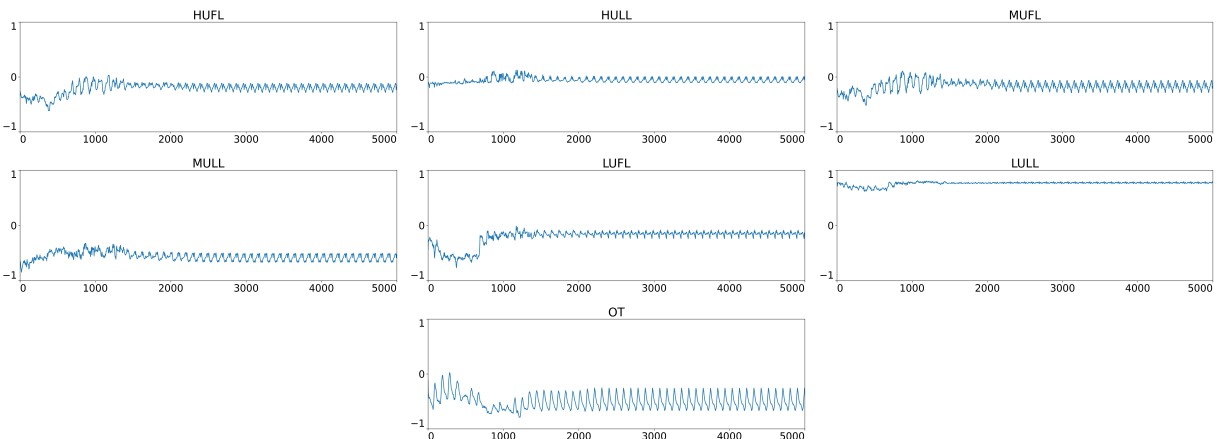

Figure 8: Example of a generated time series sample of length $l = 5000$ from the ETT dataset. The model was trained on sequences up to $l = 1000$. Up to this length, the synthesis closely follows the patterns present in the original data. Beyond this point, a stable state emerges. Most channels no longer reflect the dataset's characteristic patterns, though the "OT" channel still produces plausible structures.

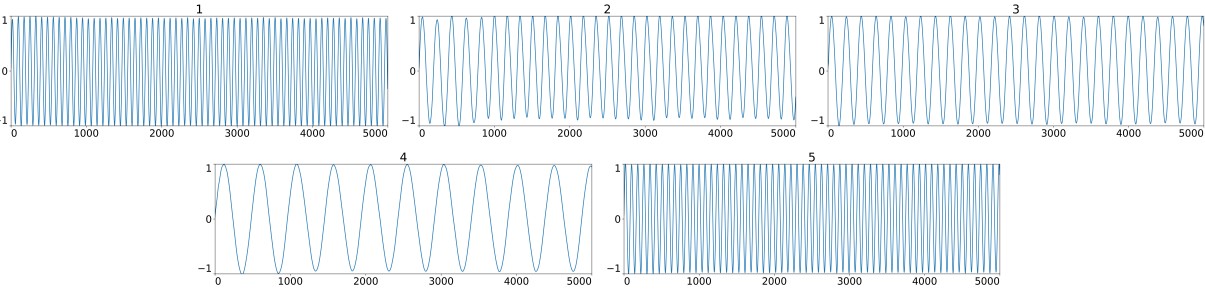

Figure 9: Example of a generated time series sample of length $l = 5000$ from the Sine dataset. The model was trained on sequences up to $l = 1000$. The sine curves are extended very consistently beyond the trained length, maintaining the dataset's structure. Upon closer inspection, a slight decrease in amplitude can be observed in channels 2 and 4 compared to the initial segment (up to $l = 1000$).

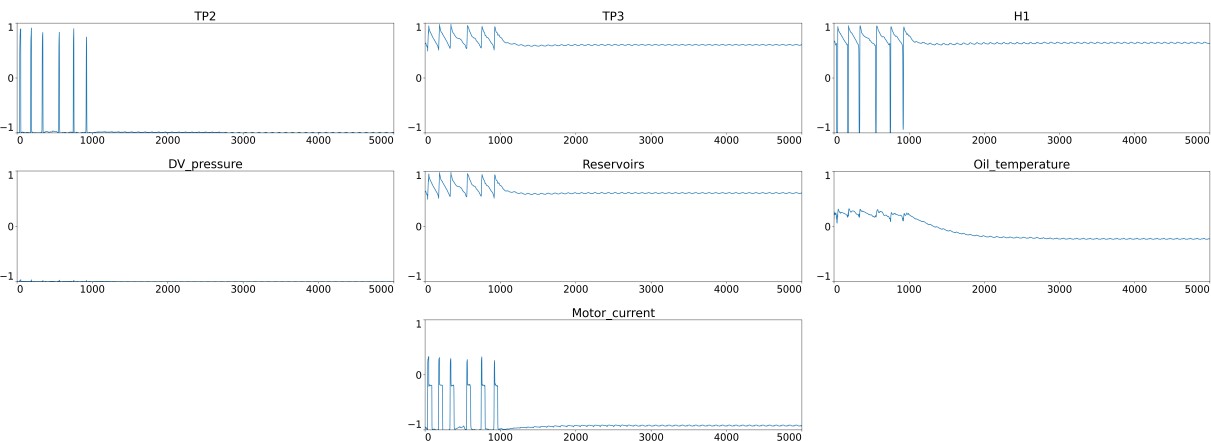

Figure 10: Example of a generated time series sample of length $l = 5000$ from the MetroPT3 dataset. The model was trained on sequences up to $l = 1000$. While the generation follows the original data up to this length, no meaningful structure is preserved in the extended part. Still, a stable state emerges, with the model settling into repetitive, low-variation patterns resembling noisy flatlines across all channels. This behavior is expected, as the MetroPT3 dataset exhibits low quasi-periodicity.

## A.3 Ablation: Decoder inductive bias

To disentangle the effect of recurrence from the effect of our decoder inductive bias, we compare AEQ-RVAE-ST against a control decoder that keeps the same recurrent backbone but removes the key constraints of our design (length-independent parameterization and approximate time-shift equivariance). We train this control variant on all datasets and sequence lengths used in the main paper ($l \in \{100, 300, 500, 1000\}$) and report FID scores. For both decoders, we annotate each layer with its input and output dimensions to make the data flow explicit.

**AEQ-RVAE-ST decoder (repeat-vector + shared per-time-step projection).** For reference, the AEQ-RVAE-ST decoder broadcasts the latent code $z$ to all time steps via a RepeatVector, decodes with a stack of LSTMs, and applies the same linear output projection independently at each time step:

```
z                                       # R^{20}
RepeatVector(n=l)                       # R^{20} -> R^{l x 20}
LSTM(256, return_sequences=True) x 4    # R^{l x 20} -> R^{l x 256}
TimeDistributed(Dense(d_c))             # R^{l x 256} -> R^{l x d_c}
```

where $d_z = 20$ is the latent dimension, $d_h = 256$ the LSTM hidden dimension, and $d_c$ denotes the number of channels (dataset-dependent). This design is length-independent: no layer's parameterization depends on $l$, since the RepeatVector merely copies $z$ along the time axis and the TimeDistributed layer applies the same weight matrix $W \in \mathbb{R}^{256 \times d_c}$ at every time step.

**Control decoder (recurrent, but without equivariance/length-independence constraints).** The control decoder maps $z$ through a dense layer and reshapes it to a length-$l$ sequence with 256 features per time step, followed by the same four-layer LSTM stack. In contrast to AEQ-RVAE-ST, the output is produced via a global projection that flattens all recurrent states and predicts the full sequence jointly:

```
z                                       # R^{20}
Dense(l*256, relu)                      # R^{20} -> R^{l*256}
Reshape((l, 256))                       # R^{l*256} -> R^{l x 256}
LSTM(256, return_sequences=True) x 4    # R^{l x 256} -> R^{l x 256}
Flatten()                               # R^{l x 256} -> R^{l*256}
Dense(l*d_c)                            # R^{l*256} -> R^{l*d_c}
```

```
Reshape((l, d_c))                                    # R^{l*d_c} -> R^{l x d_c}
```

After flattening, the final dense layer has a weight matrix $W' \in \mathbb{R}^{(l \cdot 256) \times (l \cdot d_c)}$, which can implement position-specific (absolute-time-dependent) mappings. Its parameterization explicitly depends on the target length $l$. This makes it a suitable control: it preserves recurrence, but removes the specific decoder structure that enforces our intended inductive bias.

Table 5 reports FID scores (lower is better) for AEQ-RVAE-ST and the RVAE control decoder across all datasets and sequence lengths. The control decoder can produce reasonable results on short horizons (notably on ECG at $l = 100$, where both models are comparable within confidence intervals), but its performance degrades strongly as the sequence length increases. This degradation is particularly pronounced on ETT, Sine and MetroPT3 at $l = 1000$, while AEQ-RVAE-ST remains substantially more stable. Overall, these results suggest that the decoder constraints in AEQ-RVAE-ST become increasingly important for maintaining sample quality on longer horizons.

Table 5: **FID score** of synthetic time series for the decoder ablation, comparing AEQ-RVAE-ST (ours; repeat-vector + time-distributed output) against the RVAE baseline decoder (standard LSTM-VAE style decoder). Scores are computed on the five datasets (see 4.1) at sequence lengths $l = 100$, $l = 300$, $l = 500$, and $l = 1000$. Lower scores indicate better performance. Each score is based on 5000 generated samples; results are averaged over 15 independently trained models and reported with 1-sigma confidence intervals.

| Dataset | Model | Sequence lengths | | | |
| | | 100 | 300 | 500 | 1000 |
|---|---|---|---|---|---|
| Electric Motor | **AEQ-RVAE-ST (ours)** | **0.35±0.04** | **0.12±0.01** | **0.10±0.01** | **0.24±0.02** |
| | RVAE Control | 0.81±0.09 | 0.74±0.07 | 1.44±0.15 | 1.65±0.11 |
| ECG | **AEQ-RVAE-ST (ours)** | 0.08±0.01 | **0.09±0.02** | **0.14±0.02** | **0.46±0.06** |
| | RVAE Control | **0.07±0.01** | 0.39±0.03 | 0.65±0.06 | 2.04±0.15 |
| ETT | **AEQ-RVAE-ST (ours)** | **0.58±0.05** | **0.65±0.07** | **0.79±0.07** | **1.82±0.16** |
| | RVAE Control | 0.65±0.05 | 1.13±0.09 | 2.56±0.21 | 7.97±0.56 |
| Sine | **AEQ-RVAE-ST (ours)** | **0.33±0.04** | **0.34±0.02** | **0.46±0.03** | **0.42±0.03** |
| | RVAE Control | 1.37±0.14 | 1.52±0.13 | 9.93±0.84 | 11.5±0.49 |
| MetroPT3 | **AEQ-RVAE-ST (ours)** | **0.26±0.04** | **0.65±0.07** | 2.81±0.37 | **2.84±0.22** |
| | RVAE Control | 0.60±0.06 | 1.10±0.11 | **2.34±0.27** | 13.31±0.91 |

## A.4 Power Spectral Density Analysis

We use power spectral density (PSD) analysis to evaluate how well AEQ-RVAE-ST captures frequency characteristics. Specifically, we ask two questions: (1) Does the model learn to generate time series with the correct PSD? (2) Does it hallucinate quasi-periodic structures that are not present in the training data?

The training data consists of 30,000 synthetic sequences sampled from a target PSD with two closely spaced Gaussian peaks at $f_1 = 0.08$ and $f_2 = 0.12$ (normalized frequency). The peaks have amplitudes $A_1 = 1.0$ and $A_2 = 0.9$, width $\sigma = 0.002$, and sit on a noise floor of $\epsilon = 10^{-4}$. To generate each sequence, we compute the magnitude spectrum from the target PSD, add uniformly random phases, and apply an inverse FFT. We then min-max scale the entire dataset to $[-1, 1]$. Figures 11 and 12 show the PSD and example samples for both the original training data and AEQ-RVAE-ST-generated data across sequence lengths $l \in \{100, 500, 900, 1000\}$.

For $l = 100$, the generated PSD shows oscillatory artifacts at frequencies above $f_2$. These oscillations decay in absolute value with increasing frequency, but their amplitude remains constant. The model also suppresses spectral content below $f_1$, removing the noise floor $\epsilon$. At $l = 500$, the oscillatory artifacts become stronger

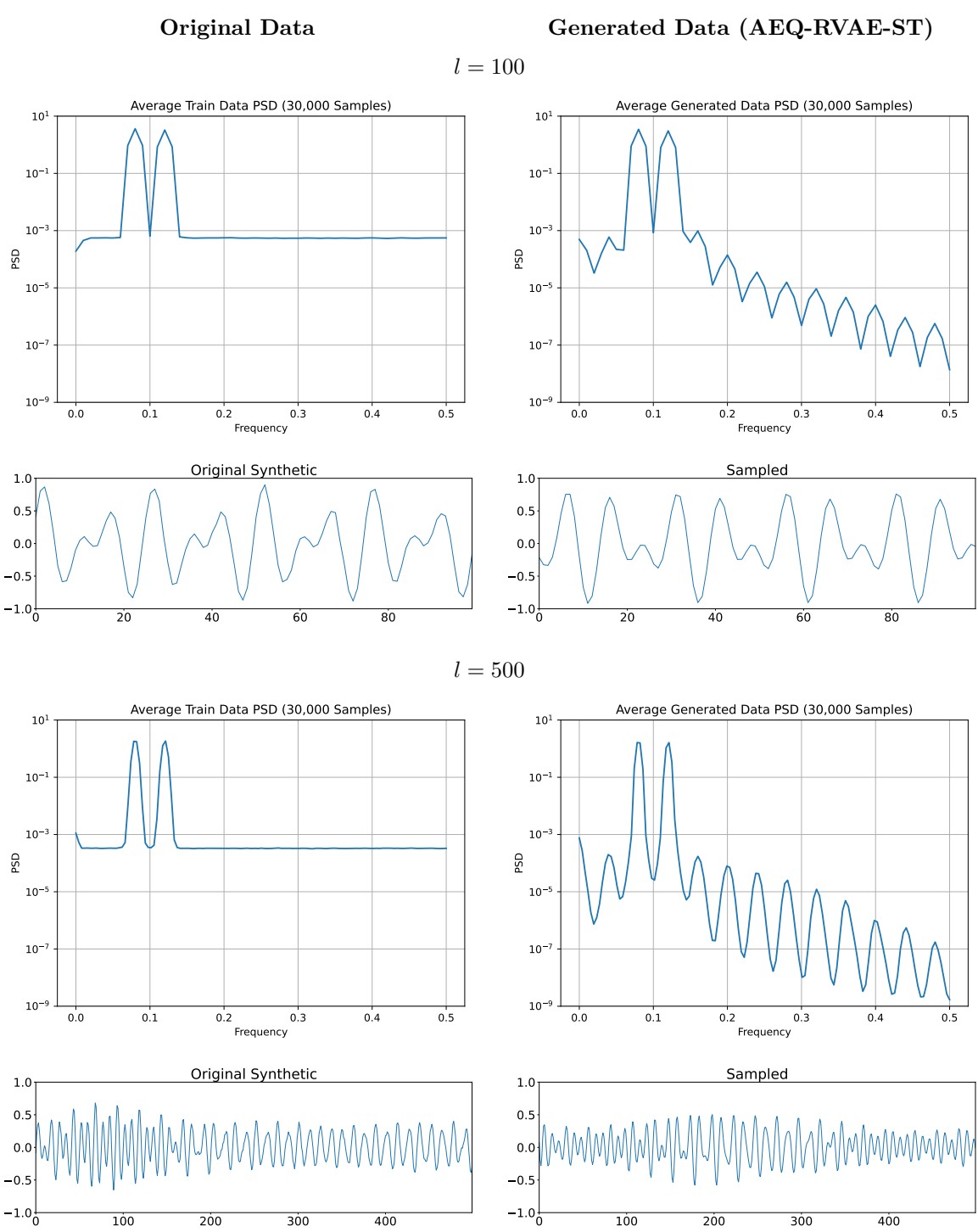

Figure 11: PSD and sample comparison for $l = 100$ and $l = 500$. Top row per sequence length: PSD. Bottom row: example samples.

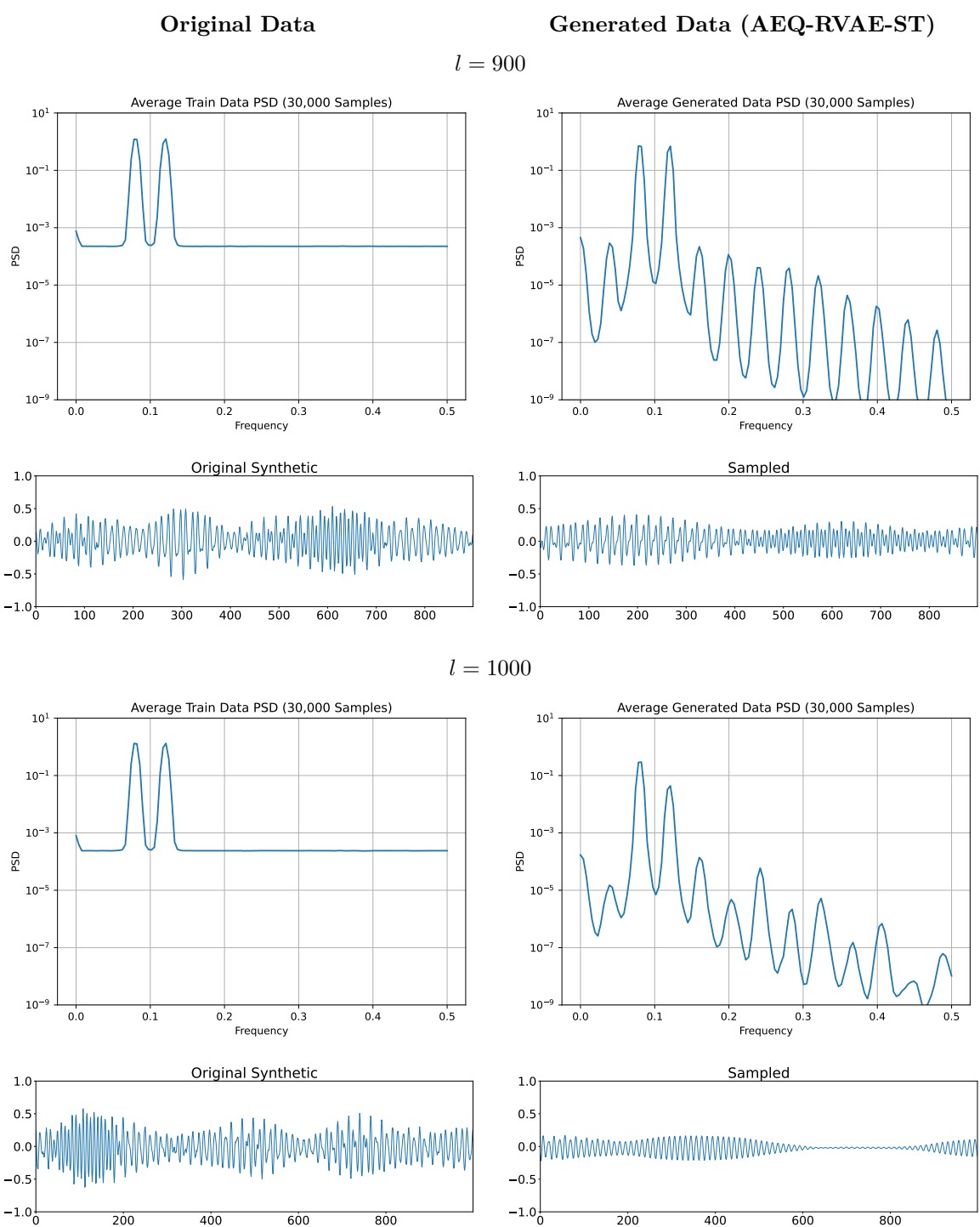

Figure 12: PSD and sample comparison for $l = 900$ and $l = 1000$. Top row per sequence length: PSD. Bottom row: example samples.

and now also appear below $f_1$. The spectral content between $f_1$ and $f_2$ is clearly reduced compared to the original, showing that the model isolates the two dominant peaks while filtering out surrounding frequencies. At $l = 900$, these effects become more pronounced: non-dominant frequencies are further suppressed and the oscillatory artifacts are stronger across the spectrum. At $l = 1000$, something different happens: $f_2$ becomes weaker relative to $f_1$, indicating that the model has difficulty preserving both frequency components at this sequence length. The oscillatory artifacts at high frequencies also change character and no longer maintain constant amplitude.

The visual inspection of the samples confirms these findings. At $l = 100$, the generated sample looks nearly identical to the original. At $l = 500$, one can see slight smoothing in the high-frequency components upon close inspection, matching the noise floor suppression in the PSD. At $l = 900$, the generated samples begin to show slightly compressed peaks compared to the originals. At $l = 1000$, this compression becomes clearly visible: both frequencies are still present, but the amplitude range of the generated samples is noticeably reduced.

To summarize: AEQ-RVAE-ST learns to generate time series that approximately match the target PSD, successfully capturing both dominant frequencies up to $l = 900$, though increasingly filtering out weaker spectral content. At $l = 1000$, the model struggles to preserve the relative amplitude of both peaks. We observe no hallucinated quasi-periodic structures—the oscillatory artifacts in the PSD represent spectral leakage rather than spurious periodicities and do not appear as visible patterns in the generated samples.

### A.5 Wall-clock training time comparison

Table 6: **Training time** (wall-clock) comparison for the evaluated models on the Electric Motor dataset at sequence length $l = 1000$. For Diffusion-TS we report training time and (separately) the time to sample 10,000 sequences.

| Model | Training time | Notes |
|---|---|---|
| AEQ-RVAE-ST (subsequent train) | 8 h | progressive training with +100 length increments |
| AEQ-RVAE-ST (no subsequent) | 2 h | standard training |
| Time-Transformer | 1 h 20 min | |
| TimeVAE | 11 min | |
| Diffusion-TS | 2 h 30 min + 1 h | +1 h for sampling 10,000 samples |
| WaveGAN | 40 min | |
| TimeGAN | 20 d | |

Table 6 reports indicative wall-clock training times on the Electric Motor dataset at sequence length $l = 1000$. These measurements are intended as order-of-magnitude estimates of computational cost and may vary with implementation details and system load; moreover, the runs were not performed on identical hardware.

Specifically, AEQ-RVAE-ST, TimeVAE, and WaveGAN were trained on a workstation (Ryzen 9 5950X, NVIDIA RTX 3080), while the remaining models were trained on a DGX system (EPYC 7742, NVIDIA A100). For Diffusion-TS, we additionally report the time required to generate 10,000 synthetic sequences after training.

For AEQ-RVAE-ST, we distinguish standard training from subsequent training. In subsequent training, the sequence length is increased progressively (from $l = 100$ to $l = 1000$ in increments of 100), which increases overall runtime compared to training directly at $l = 1000$.

Overall, the results suggest clear runtime differences in our setup. The fastest models are the convolution-based TimeVAE and WaveGAN, whereas the recurrent models (AEQ-RVAE-ST and TimeGAN) require substantially longer training times. In addition, diffusion-based models require additional time for sample generation after training.

## A.6 Hyperparameters and Loss Function

In all experiments, for the encoder as well as the decoder, we stack 4 LSTM-layers each with 256 hidden units. The latent dimension is $z = 20$. We use Adam optimizer with learning rate $\alpha = 10^{-4}$, $\beta_1 = 0.9$, $\beta_2 = 0.999$, $\epsilon = 10^{-7}$. We perform min-max scaling with $(-1, 1)$. After scaling we do a train/validation split with a ratio of 9:1.

We use the loss function

$$\mathcal{L}_{\theta,\phi} = \alpha \cdot \text{SSE} + \beta \cdot \text{D}_{\text{KL}}, \tag{5}$$

where the reconstruction loss, SSE, represents the sum of squared errors, computed for each individual sample within a batch:

$$\text{SSE} = \sum_T \sum_C (y_{tc} - \hat{y}_{tc})^2, \tag{6}$$

where $T$ is the sequence length and $C$ is the number of channels. We then average the SSE over the entire batch. In our experiments we set $\alpha = \frac{500}{T}$ and $\beta = 0.1$.

The parameter $\beta$ follows the $\beta$-VAE framework (Higgins et al., 2017): for $0 < \beta < 1$ the reconstructed samples are less smoothed, while $\beta > 1$ encourages disentangled representations (Burgess et al., 2018). We adjust $\alpha$ antiproportional to the sequence length to retain the ratio between the reconstruction loss and the KL-Divergence.

## A.7 Loss to ELBO conversion

Given the Gaussian likelihood with variance $\sigma^2$, the log-likelihood can be expressed in terms of the SSE:

$$\text{SSE} = -2\sigma^2 \cdot \text{log-likelihood} - \sigma^2 \log\left(2\pi\sigma^2\right) \cdot T \cdot C. \tag{7}$$

Using $\text{ELBO} = \text{log-likelihood} - \text{D}_{\text{KL}}$ (Murphy, 2022), together with (5), (7) and $\sigma^2 = 0.5 \cdot \frac{\beta}{\alpha}$, we derive:

$$\frac{\mathcal{L}_{\theta,\phi}}{\beta} = \frac{\alpha}{\beta} \cdot \text{SSE} + \text{D}_{\text{KL}}$$

$$= -\text{log-likelihood} - 0.5 \cdot \log\left(\pi \cdot \frac{\beta}{\alpha}\right) \cdot T \cdot C + \text{D}_{\text{KL}}$$

$$\implies \text{ELBO}(\mathcal{L}_{\theta,\phi}, \alpha, \beta, T, C) = -\frac{\mathcal{L}_{\theta,\phi}}{\beta} - 0.5 \cdot \log\left(\pi \cdot \frac{\beta}{\alpha}\right) \cdot T \cdot C. \tag{8}$$

We normalize the ELBO by the product of sequence length and number of channels to enable comparison across datasets with different dimensionalities:

$$\text{ELBO}_{\text{norm}}(\mathcal{L}_{\theta,\phi}, \alpha, \beta, T, C) = \frac{\text{ELBO}(\mathcal{L}_{\theta,\phi}, \alpha, \beta, T, C)}{T \cdot C}. \tag{9}$$

## A.8 Evaluation by Average ELBO

For completeness, we evaluate the average Evidence Lower Bound (ELBO) on a synthetic dataset $\tilde{X} \in \mathbb{R}^{n_s \times l \times c}$ where $n_s$ represents the numbers of samples, $l$ denotes the sequence length, and $c$ the number of channels. We refer to this metric as $\mathcal{E}_{\text{avg}}(\tilde{X})$. In detail, we first train a VAE model on shorter sequence lengths $\ell \ll l$, which facilitates easier training. Since this metric reflects short-term reconstruction quality only, it is not used for model ranking in our main evaluation.

We then calculate the *average ELBO*:

$$\mathcal{E}_{\text{avg}}(\tilde{X}) = \frac{1}{n_s(l-\ell)} \sum_{i=0}^{n_s-1} \sum_{t=0}^{l-\ell-1} \text{ELBO}_{\text{norm}}\left(\mathcal{L}_{\theta,\phi}(\tilde{X}_{i,t:t+\ell,\cdot})\right), \tag{10}$$

Table 7: **Average *ELBO* score** $\mathcal{E}_{\text{avg}}(\tilde{X})$ of synthetic time series for six models (see 4.2), computed on the five datasets (see 4.1) at sequence lengths $l = 100$, $l = 300$, $l = 500$, and $l = 1000$. Higher scores indicate better performance. Each score is based on 1000 generated samples evaluated with an *ELBO model* using the AEQ-RVAE-ST architecture, with 1-sigma confidence intervals. Note that while the ELBO score is generally informative, it can overestimate quality on certain datasets such as Sine and ECG, where implausible outputs may go undetected. For the Sine dataset in particular, uncorrelated channels and high sensitivity to local artifacts limit the reliability of this metric. [†] Overestimated due to local consistency effects (flat lines).

| Dataset | Model | Sequence lengths | | | |
|---|---|---|---|---|---|
| | | 100 | 300 | 500 | 1000 |
| Electric Motor | **AEQ-RVAE-ST(ours)** | **1.62±0.69** | **1.65±0.60** | **1.66±0.03** | **1.65±0.03** |
| | TimeGAN | 1.20±0.59 | 1.33±0.48 | 1.13±0.56 | -4.05±2.41 |
| | WaveGAN | 1.54±0.11 | 1.54±0.16 | 1.54±0.14 | 1.53±0.37 |
| | TimeVAE | 1.49±0.88 | 1.38±1.34 | 1.09±2.21 | 0.31±3.24 |
| | Diffusion-TS | 1.58±0.06 | 1.36±0.26 | 1.38±0.24 | 1.30±0.25 |
| | Time-Transformer | 0.98±2.46 | -28.9±3.33 | -21.7±0.91 | -28.4±4.12 |
| ECG | **AEQ-RVAE-ST(ours)** | **1.64±0.13** | **1.64±0.18** | **1.63±0.20** | **1.59±0.27** |
| | TimeGAN | -14.6±1.87 | -14.6±1.41 | -13.7±6.67 | -15.3±2.57 |
| | WaveGAN | 1.12±0.81 | 1.11±0.87 | 1.10±0.86 | 1.10±0.83 |
| | TimeVAE | 1.55±0.37 | 1.37±0.65 | 1.26±0.70 | 0.87±0.92 |
| | Diffusion-TS | **1.65±0.07** | **1.64±0.19** | 1.60±0.29 | 1.29±1.00 |
| | Time-Transformer | 1.07±0.85[†] | 1.68±0.05[†] | 1.68±0.05[†] | 1.68±0.05[†] |
| ETT | **AEQ-RVAE-ST(ours)** | **1.49±0.52** | **1.50±0.40** | **1.52±0.35** | **1.53±0.63** |
| | TimeGAN | 1.39±0.70 | 0.85±3.36 | -4.29±9.66 | -0.38±0.65 |
| | WaveGAN | 1.40±0.53 | 1.39±0.70 | 1.42±0.51 | 1.42±0.48 |
| | TimeVAE | 1.47±0.94 | 1.20±1.54 | 0.89±1.99 | 0.42±2.45 |
| | Diffusion-TS | **1.50±0.18** | 1.49±0.26 | 1.50±0.27 | 1.50±0.17 |
| | Time-Transformer | 1.07±1.93[†] | 1.38±0.86[†] | 1.49±0.14[†] | -39.9±5.84[†] |
| Sine | AEQ-RVAE-ST(ours) | 1.42±0.25 | **1.19±0.55** | **1.28±0.48** | **1.41±0.27** |
| | TimeGAN | -0.59±2.47 | -1.25±2.72 | -2.33±3.21 | -4.73±5.64 |
| | WaveGAN | -1.28±2.20 | -1.04±1.76 | -0.97±1.72 | -0.94±1.75 |
| | TimeVAE | 1.06±0.66 | -3.55±8.18 | -6.21±9.38 | -8.81±12.1 |
| | Diffusion-TS | **1.50±0.06** | 1.14±0.53 | 0.56±1.09 | -0.30±1.60 |
| | Time-Transformer | 1.23±0.49[†] | -0.12±1.45[†] | 1.18±0.87[†] | 1.34±0.65[†] |
| MetroPT3 | **AEQ-RVAE-ST(ours)** | 1.41±1.74 | 0.76±3.49 | 0.57±3.78 | 0.60±3.75 |
| | TimeGAN | 1.25±1.38 | 0.61±4.39[†] | 1.46±1.36[†] | -11.1±18.2[†] |
| | WaveGAN | -1.71±4.85 | -1.62±4.90 | -1.64±4.83 | -1.68±4.91 |
| | TimeVAE | -0.07±3.96 | -2.06±5.91 | -5.64±7.29 | -9.03±7.38 |
| | Diffusion-TS | **1.63±0.92** | **1.43±2.21** | **1.36±2.53** | **0.77±3.50** |
| | Time-Transformer | -2.30±5.81 | -3.05±6.55 | -2.97±0.55 | -302±14.4 |

where $\mathcal{L}_{\theta,\phi}$ is the loss of the *ELBO Model* and $\text{ELBO}_{\text{norm}} = \text{ELBO} \cdot (cl)^{-1}$ is a normalized ELBO, as explained in Appendix A.7. By normalizing the ELBO, we get a fairer comparison of datasets with different dimensionalities and varying sequence lengths.

$\mathcal{E}_{\text{avg}}(\tilde{X})$ gives us information about short term consistency over the whole synthetic dataset. We chose $\ell = 50$ which is half of the lowest sequence length in the experiments. A well trained *ELBO model* (An & Cho, 2015) allows us to evaluate the (relative) short term consistency of synthetic data in high accuracy and low variance. To ensure reliable assessment of sample quality, we prevented overfitting of the *ELBO*

Table 8: **Average *ELBO* score** $\mathcal{E}(\tilde{X})$ of synthetic time series for six models (see 4.2), computed on the five datasets (see 4.1) at sequence lengths of $l = 100$, $l = 300$, $l = 500$, and $l = 1000$. A higher score indicates better performance. For each score, 1000 generated samples were evaluated by an *ELBO model* (based on the TimeVAE architecture) and the results are reported with 1-sigma confidence intervals. The interpretation must follow analogously to the explanation provided in Section A.8 of the main paper, where the specifics and limitations of the ELBO score are discussed in detail. [†] Overestimated due to local consistency effects (flat lines).

| Dataset | Model | Sequence lengths | | | |
| | | 100 | 300 | 500 | 1000 |
|---|---|---|---|---|---|
| Electric Motor | **AEQ-RVAE-ST (ours)** | **1.61±0.69** | **1.64±0.12** | **1.64±0.01** | **1.64±0.02** |
| | TimeGAN | 1.29±0.39 | 1.33±0.17 | 1.21±0.10 | -2.14±0.82 |
| | WaveGAN | 1.52±0.14 | 1.47±1.05 | 1.52±0.22 | 1.52±0.15 |
| | TimeVAE | 1.52±0.87 | 1.44±1.28 | 1.01±2.35 | 0.10±3.58 |
| | Diffusion-TS | 1.56±0.45 | 1.35±0.36 | 1.39±0.21 | 1.30±0.29 |
| | Time-Transformer | 1.25±1.88 | -22.9±7.52 | -85.4±18161 | -22.7±8.05 |
| ECG | **AEQ-RVAE-ST (ours)** | 1.62±0.07 | 1.62±0.07 | **1.62±0.06** | **1.59±0.06** |
| | TimeGAN | -2.57±0.22 | -2.26±0.22 | -2.67±1.92 | -2.58±0.49 |
| | WaveGAN | 1.32±0.29 | 1.33±0.18 | 1.32±0.16 | 1.32±0.15 |
| | TimeVAE | 1.57±0.15 | 1.46±0.16 | 1.39±0.15 | 1.08±0.28 |
| | Diffusion-TS | **1.63±0.06** | **1.63±0.08** | 1.60±0.18 | 1.16±25.2 |
| | Time-Transformer | 1.22±0.50 | 1.67±0.04[†] | 1.67±0.04[†] | 1.67±0.04[†] |
| ETT | **AEQ-RVAE-ST (ours)** | **1.56±0.24** | **1.57±0.09** | **1.59±0.05** | **1.60±0.13** |
| | TimeGAN | 1.49±0.17 | 1.20±1.49 | 0.83±0.91 | -0.00±0.28 |
| | WaveGAN | 1.50±0.50 | 1.50±0.41 | 1.47±0.64 | 1.49±0.43 |
| | TimeVAE | **1.56±0.45** | 1.41±0.81 | 1.15±1.05 | 0.40±2.06 |
| | Diffusion-TS | 1.53±0.07 | 1.52±0.13 | 1.52±0.13 | 1.52±0.16 |
| | Time-Transformer | 1.43±0.52 | 1.57±0.11[†] | 1.48±0.04[†] | -39.6±5.63 |
| Sine | AEQ-RVAE-ST(ours) | 1.46±0.07 | **1.44±0.09** | **1.45±0.06** | **1.47±0.04** |
| | TimeGAN | 0.66±1.04 | 0.39±1.18 | -0.19±1.59 | -2.16±3.70 |
| | WaveGAN | 0.29±0.86 | 0.50±0.70 | 0.55±0.66 | 0.60±0.66 |
| | TimeVAE | 1.42±0.12 | 0.66±2.38 | 0.04±3.04 | -0.81±4.20 |
| | Diffusion-TS | **1.48±0.02** | **1.44±0.10** | 1.33±0.16 | 1.23±0.19 |
| | Time-Transformer | 1.44±0.09[†] | 1.27±0.22[†] | 1.44±0.13[†] | 1.46±0.10[†] |
| MetroPT3 | AEQ-RVAE-ST (ours) | 1.49±0.64 | 1.38±0.77 | 1.39±0.74 | **1.36±0.81** |
| | TimeGAN | 1.33±0.77 | 0.95±1.83[†] | 1.42±0.84[†] | -0.07±2.94[†] |
| | WaveGAN | 0.35±1.57 | 0.18±1.67 | 0.23±1.64 | 0.22±1.64 |
| | TimeVAE | 1.06±1.14 | -0.07±2.07 | -2.81±3.43 | -5.61±3.36 |
| | Diffusion-TS | **1.63±0.24** | **1.58±0.49** | **1.59±0.41** | 1.04±2.08 |
| | Time-Transformer | 0.06±1.73 | -0.97±2.21 | -1.29±0.63 | -331±26.2 |

*model* by applying early stopping after 50 epochs without improvement and restoring the best weights. In our experiments, we employed two distinct *ELBO models* for calculating $\mathcal{E}_{\text{avg}}(\tilde{X})$. The first model is based on the AEQ-RVAE-ST architecture, while the second utilizes the TimeVAE framework (Desai et al., 2021a). The use of a TimeVAE-based *ELBO model* provides an additional evaluation to ensure that the AEQ-RVAE-ST-based model is not biased toward our own generated samples. As detailed in Appendix A.9, the results obtained using TimeVAE are highly similar to those produced by the AEQ-RVAE-ST-based model.

The average ELBO measures short-term consistency on subwindows of length $\ell$ and can therefore overestimate models that reproduce local statistics while failing to capture global dynamics. This effect is visible for the Time-Transformer on Sine, ECG and ETT datasets and also on for TimeGAN on the MetroPT3 dataset: Both models produce flat segments that, when evaluated on short windows, appear locally consistent with the training data and therefore inflate $\mathcal{E}_{\mathrm{avg}}$, yet they do not reflect the characteristic dynamics of the dataset. The mismatch is evident in our other scores and in the PCA and t-SNE embeddings, where these samples cluster away from the real data. Interpreted with this caveat, AEQ-RVAE-ST produces the best samples on all datasets starting at $l = 300$, with the exception of MetroPT3, where Diffusion-TS is performing best.

## A.9   Average Elbo with TimeVAE Elbo-Model

Table 8 shows the results for the average *ELBO* score $\mathcal{E}(\tilde{X})$ using the base of TimeVAE as the *ELBO model*. However, instead of using the original loss function of TimeVAE, we utilized the loss function of AEQ-RVAE-ST as it simplifies the conversion to the *ELBO* score as shown in (8). Analogous to Table 7, our model is outperforming all other models from $l = 300$ with the exception of the ECG dataset where our model is outperforming from $l = 500$. On the other side, our model is outperforming Diffusion-TS on the MetroPT3 dataset on $l = 1000$.

## A.10   Baseline Model Details

This section provides detailed descriptions of the baseline models, including their architectural properties and equivariance characteristics. Implementation details and hyperparameters for each model are provided in Appendix A.11.1.

**TimeGAN**(Yoon et al., 2019)**:** A GAN-based model that is considered state-of-the-art in generation of times series data. TimeGAN's generator has a recurrent structure like AEQ-RVAE-ST. A key difference is that it's latent dimension is equal to the sequence length. Notably, equivariance on this model is lost on the output layer of the generator which maps all hidden states at once through a linear layer to a sequence. On its initial paper release, TimeGAN was tested and compared to other models on a small sequence length of $l = 24$.

**WaveGAN** (Donahue et al., 2019)**:** A GAN-based model developed for generation of raw audio waveforms. WaveGAN's generator is based on convolutional layers. It doesn't rely on typical audio processing techniques like spectrogram representations and is instead directly working in the time domain, making it also suitable for learning time series data. It is designed to exclusively support sequence lengths in powers of 2, specifically $2^{14}$ to $2^{16}$. Notably, WaveGAN loses it's equivariance on a dense layer between the latent dimension and the generator, however the generator itself completely maintains equivariance with its upscaling approach. In our experiments, it was trained with the lowest possible sequence length of $2^{14}$, and the generated samples were subsequently split to match the required sequence length. In (Yoon et al., 2019), WaveGAN was outperformed by TimeGAN on low sequence length.

**TimeVAE** (Desai et al., 2021b)**:** A VAE-based model designed for time series generation using convolutional layers. Analogous to WaveGAN, it loses equivariance between the latent dimension and the decoder and additionally it loses equivariance on the output layer where a flattened convolutional output is passed through a linear layer. It has demonstrated performance comparable to that of TimeGAN.

**Diffusion-TS** (Yuan & Qiao, 2024)**:** A generative model for time series based on the diffusion process framework. It combines trend and seasonal decomposition with a Transformer-based architecture. A Fourier basis is used to model seasonal components, while a low-degree polynomial models trends. Samples are generated by reversing a learned noise-injection process. While the model leverages the global structure of sequences, it lacks time-translation equivariance: this is due both to the use of position embeddings in the Transformer component and to the fixed basis decomposition, which breaks shift-invariance.

**Time-Transformer** (Liu et al., 2024)**:** An adversarial autoencoder (AAE) model tailored for time series generation, integrating a novel Time-Transformer module within its decoder. The Time-Transformer employs a layer-wise parallel design, combining Temporal Convolutional Networks (TCNs) for local feature extraction and Transformers for capturing global dependencies. A bidirectional cross-attention mechanism facilitates

effective fusion of local and global features. While TCNs are inherently translation-equivariant, this property is overridden by the Transformer's positional encoding and attention structure, making the overall model not equivariant.

None of these baselines enforce approximate time-shift equivariance by design. Figure 6 illustrates that AEQ-RVAE-ST does induce this bias, which is particularly relevant for long-horizon training.

### A.11 Implementation details of baseline models

### A.11.1 Hyperparameters and model configs

To balance data diversity and computational efficiency, we used a dataset-specific step size when splitting time series into training sequences. This step size determines the offset between starting points of consecutive sequences, thereby influencing both the number of training samples and the memory requirements during training.

For the Electric Motor, ECG, and MetroPT3 datasets, we chose a step size of $0.1 \cdot l$, where $l$ is the sequence length. For the ETT dataset, which exhibits more complex and longer-range temporal dependencies, we used a smaller step size of $0.04 \cdot l$ to increase the number of training samples. In contrast, for the synthetic Sine dataset, we fixed the number of training samples to 10,000 for each sequence length.

This approach reflects a practical trade-off: while smaller step sizes increase training data diversity, they also lead to higher memory usage. Particularly for long sequences, using very small step sizes (e.g., step size = 1) can cause GPU memory overflow or even exceed system RAM, depending on the model architecture, implementation and dataset.

### A.11.2 TimeGAN

We did all experiments with the same hyperparameters. Num layers=3, hidden dim=100, num iterations = 25000. The clockwise computation time on these hyperparameters were the highest of all models. We use the authors original implementation[4] on a Nvidia DGX A100 server in the 19.12-tf1-py3 container[5]. On sequence length $l = 1000$ , the training took about 3 weeks wall-clock time.

### A.11.3 WaveGAN

For WaveGan needed special preparation to be usable for training. First we min maxed scaled the dataset file, split it into training and validation parts and then converted each into a n-dimensional *.wav* file. WaveGan is limited in configurability. In terms of sequence length the user can decide between $2^{14}$, $2^{15}$ and $2^{16}$. We chose $2^{14} = 16384$ because it is the smallest possible length. When we generate samples, we cut them into equal parts which correspond to the desired sequence length $l$. The rest of the hyperparameters were set to default. On the sine dataset training, wie created 10,000 samples with a length of 16,384. We used the ported pytorch implementation[6].

### A.11.4 TimeVAE

We use TimeVAE with default parameters. We integrated components of the original TimeVAE implementation[7], such as the encoder, decoder, and loss function, into our own program framework. The reconstruction loss of TimeVAE is

$$\sum_T \sum_C (y_{tc} - \hat{y}_{tc})^2 + \frac{1}{C} \sum_C (\bar{y}_c - \hat{\bar{y}}_c)^2. \tag{11}$$

---

[4]https://github.com/jsyoon0823/TimeGAN
[5]https://docs.nvidia.com/deeplearning/frameworks/tensorflow-release-notes/rel__19.12.html
[6]https://github.com/mostafaelaraby/wavegan-pytorch
[7]https://github.com/abudesai/timeVAE

TimeVAE includes a hyperparameter a, which acts as a weighting factor for the reconstruction loss. The authors of the original paper recommend using a value for a in the range of 0.5 to 3.5 to balance the trade-off between reconstruction accuracy and latent space regularization. In all of our experiments, we set $a = 3$.

### A.11.5 Time-Transformer

We used the official implementation[8] with default parameters. The encoder is a 3-layer 1D CNN (filters $[64, 128, 256]$, kernel size 4, dropout 0.2). The decoder uses a TimeSformer-based architecture (head size 64, 3 heads, two transposed convolution layers with filters $[128, 64]$, kernel size 4, dilations $[1, 4]$, dropout 0.2). The discriminator is an MLP with hidden dimension 32. All three components use polynomial decay learning rate schedules (power $= 0.5$, 300 steps): autoencoder $0.005 \rightarrow 0.0025$, discriminator and generator $0.001 \rightarrow 0.0001$.

### A.11.6 Diffusion-TS

We use the official implementation[9]. For all datasets except Sine, we use a unified configuration: 3 encoder layers, 2 decoder layers, model dimension $d = 64$, 500 diffusion timesteps, 4 attention heads, MLP expansion factor 4, kernel size 1, no dropout, $L1$ loss with cosine beta schedule. For the Sine dataset, we use the authors' provided configuration.

### A.12 Discriminative Score

The 2-layer RNN for binary classification consists of a GRU layer, where the hidden dimension is set to $\lfloor n_c/2 \rfloor$, where $n_c$ is the number of channels. This is followed by a linear layer with an output dimension of one. To prevent overfitting, early stopping with a patience of 50 is applied. We each discriminative score we repeated 15 training procedures. On each procedure, 2000 random samples were used as the train dataset and 500 samples were used as the validation dataset for early stopping monitoring. The discriminative score is then determined by validating further independent 500 samples.

### A.13 PyTorch vs TensorFlow

Our experiments use TensorFlow. A PyTorch reimplementation initially showed worse performance, which we traced to differing default weight initializations in LSTM and Dense layers. After aligning both frameworks to use uniform initialization, results were consistent.

### A.14 PCA and t-SNE Results

This section presents PCA and t-SNE plots for all datasets at sequence lengths $l = 100$ and $l = 1000$. The EM and ECG PCA plots shown in the main text (Figure 4) are not repeated here. Since TimeGAN and WaveGAN show consistent performance across sequence lengths within a given dataset, these observations will not be explicitly mentioned in each figure caption.

---

[8]https://github.com/Lysarthas/Time-Transformer
[9]https://github.com/Y-debug-sys/Diffusion-TS

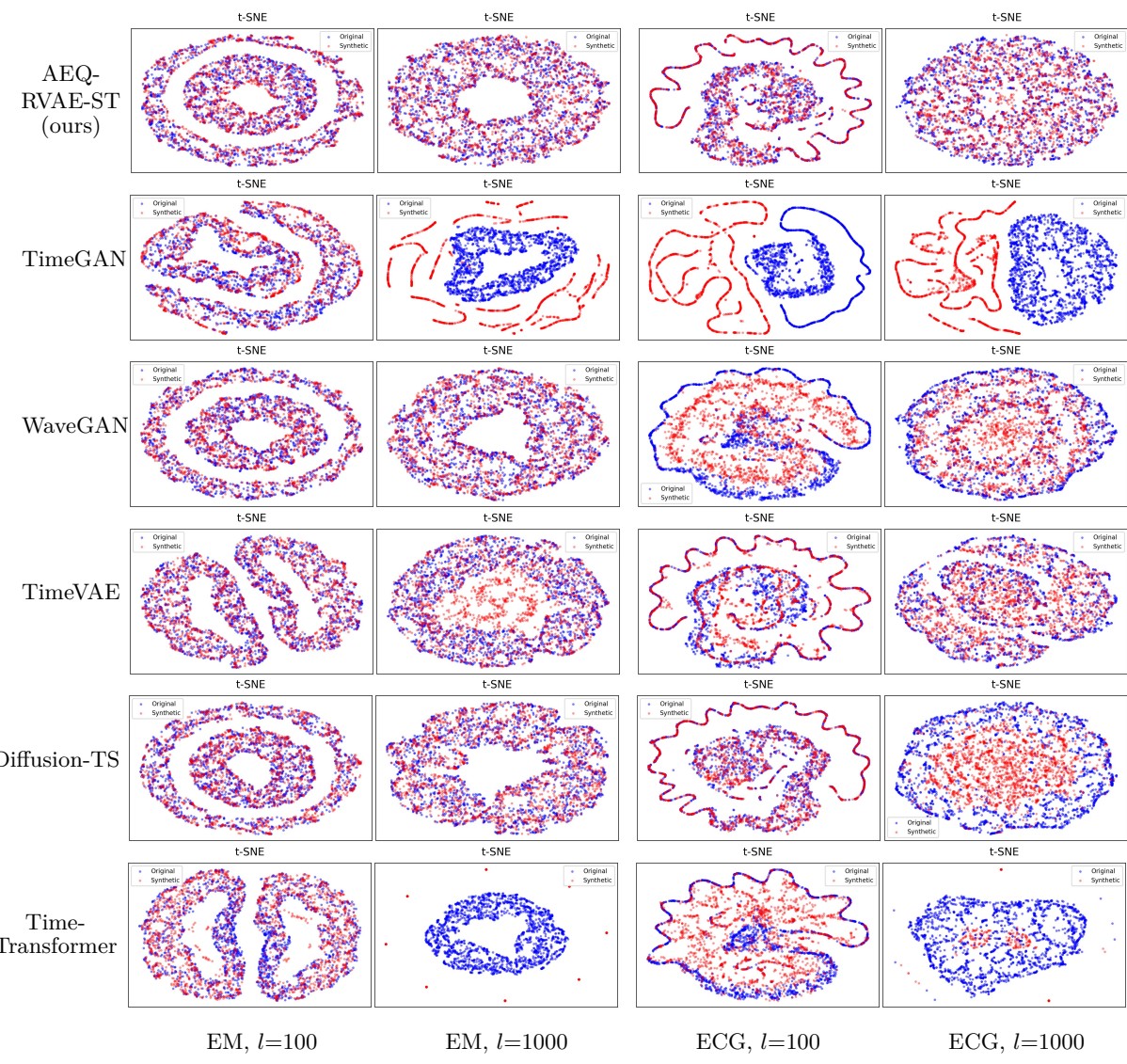

Figure 13: t-SNE plots for sequence lengths $l = 100$ and $l = 1000$ on the Electric Motor (EM) and ECG datasets. At $l = 100$, TimeGAN already performs worse than the other models on both datasets, similarly to Time-Transformer. At $l = 1000$, AEQ-RVAE-ST shows the best performance on ECG, while on EM, AEQ-RVAE-ST, WaveGAN, and Diffusion-TS perform similarly. TimeGAN and Time-Transformer fail to generate coherent samples at longer sequence lengths on both datasets.

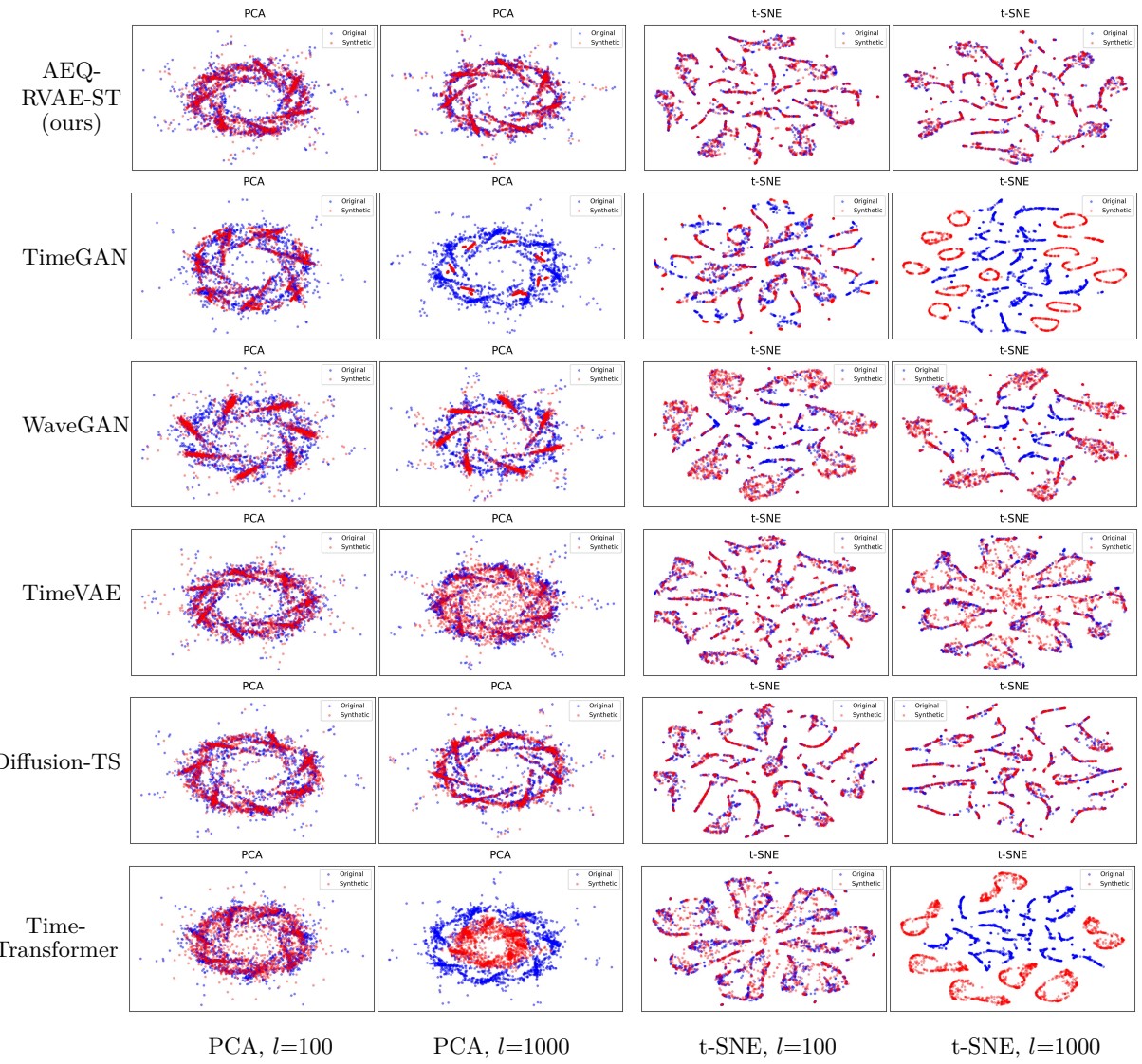

Figure 14: PCA and t-SNE plots for sequence lengths $l = 100$ and $l = 1000$ on the ETT dataset. AEQ-RVAE-ST and Diffusion-TS consistently perform the best across all sequence lengths. WaveGAN fails to capture the full variance of the dataset. TimeVAE performs similarly to AEQ-RVAE-ST and Diffusion-TS at $l = 100$, but its performance degrades with increasing sequence length. TimeGAN and Time-Transformer perform reasonably well at $l = 100$, though already worse than the other models, and their performance significantly drops at $l = 1000$.

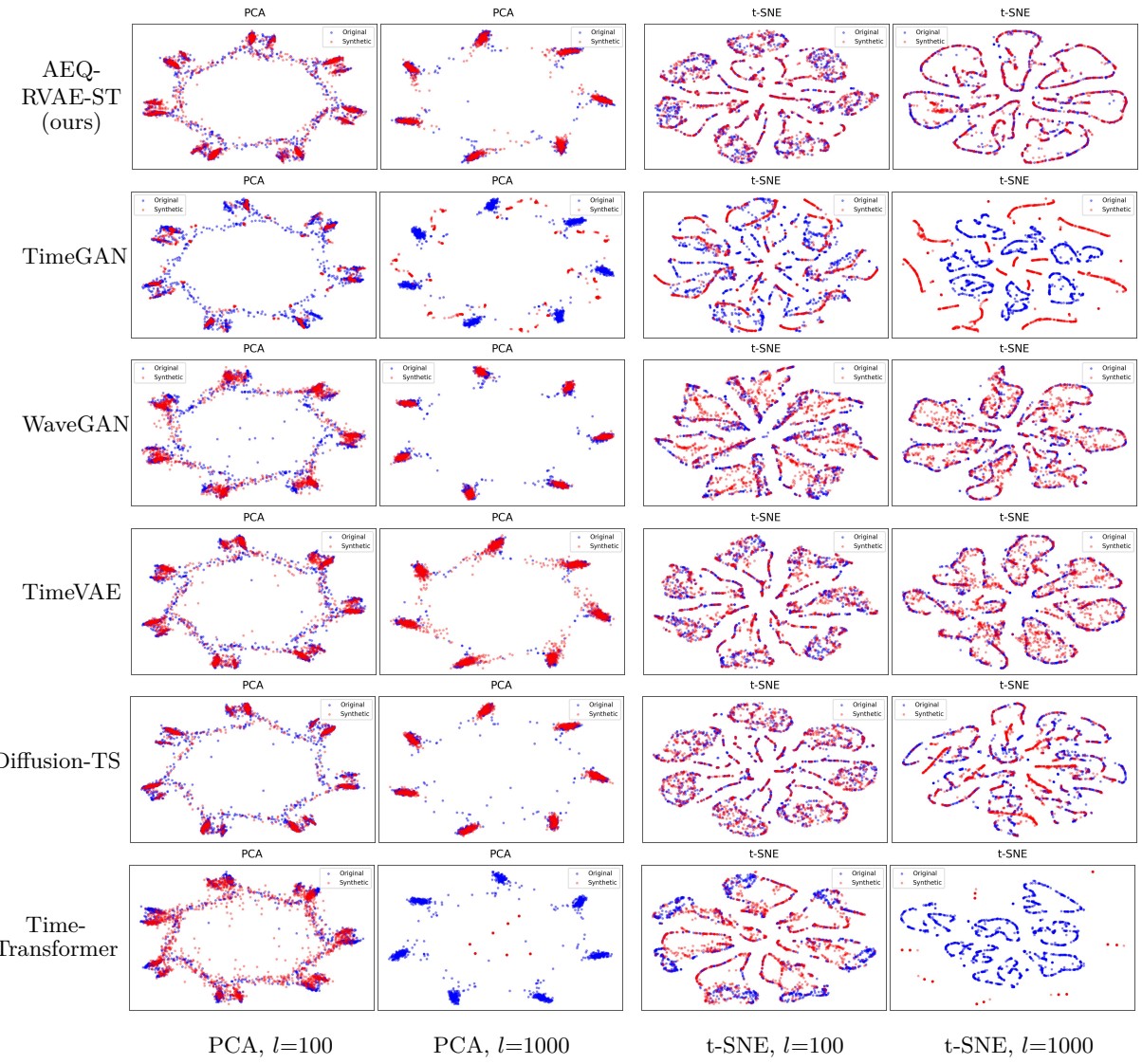

Figure 15: PCA and t-SNE plots for sequence lengths $l = 100$ and $l = 1000$ on the MetroPT3 dataset. AEQ-RVAE-ST, TimeVAE and Diffusion-TS perform similarly and the best across all sequence lengths. WaveGAN performs slightly worse, as it does not capture the entire distribution of the dataset. Time-Transformer performs reasonably well at $l = 100$, but its performance degrades at longer sequence lengths. TimeGAN consistently performs the worst, failing to generate plausible samples.

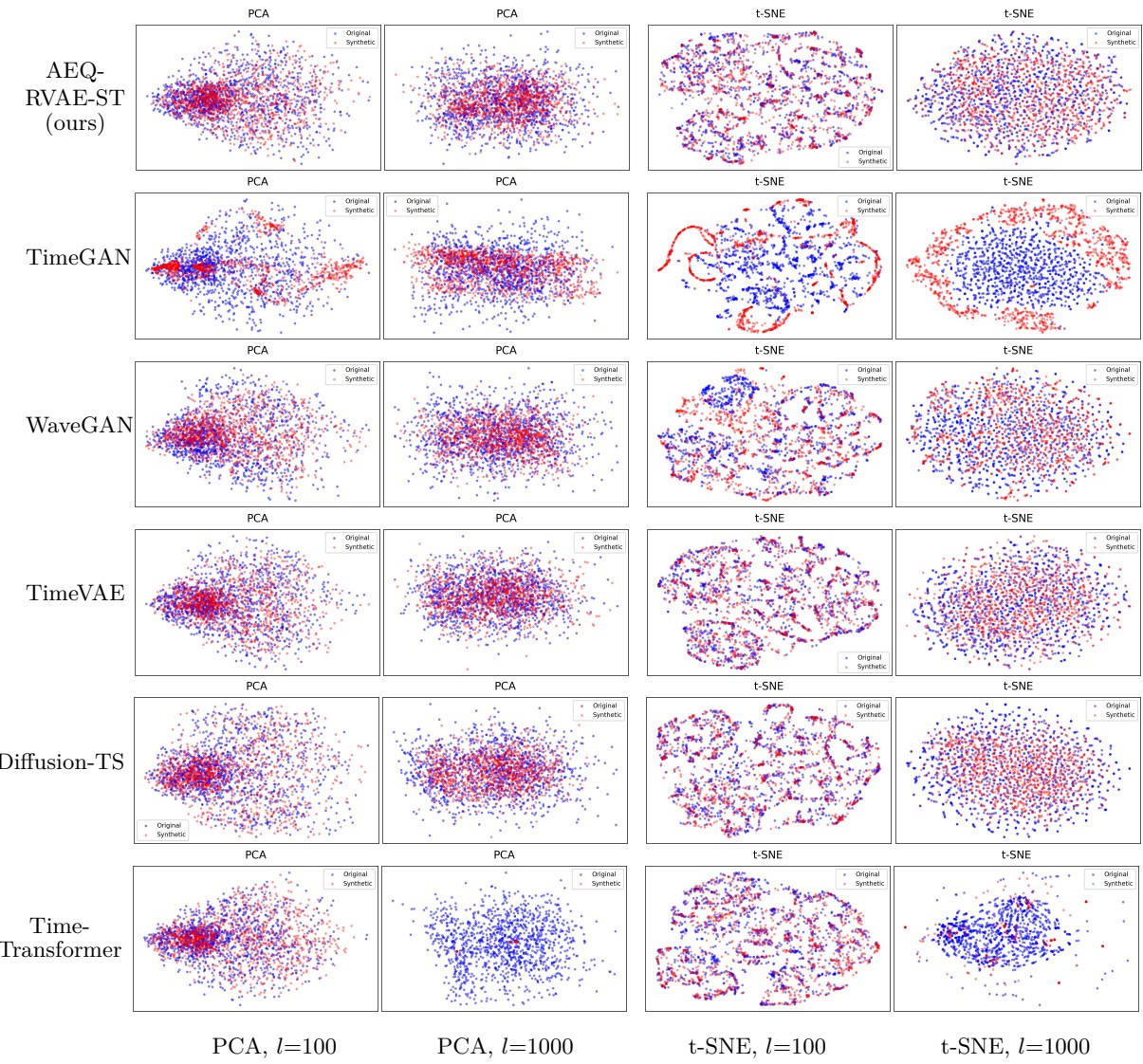

Figure 16: PCA and t-SNE plots for sequence lengths $l = 100$ and $l = 1000$ on the Sine dataset. At $l = 100$, all models perform similarly well, except TimeGAN which consistently performs less effectively. At $l = 1000$, Time-Transformer also shows a decline in performance. AEQ-RVAE-ST, Diffusion-TS, WaveGAN and TimeVAE perform equally well at both sequence lengths. However, when looking at Figure 5, this does not fully reflect the models performance, as the limitations in accounting for temporal dependencies lead to significantly reduced effectiveness.

