# OpenReview forum: "Approximately Equivariant Recurrent Generative Models for Quasi-Periodic Time Series with a Progressive Training Scheme"
_TMLR — Accepted by TMLR_

### Review · Reviewer_5aBU · 2025-12-12

**Summary Of Contributions:**

This paper proposes RVAE-ST, a recurrent variational autoencoder tailored for long, quasi-periodic time series generation. The core contribution is a principled combination of (i) an approximately time-shift-equivariant recurrent architecture, (ii) a sequence-length–independent parameterization, and (iii) a progressive training scheme that gradually increases sequence length to stabilize optimization and enable learning over long horizons.

Conceptually, the paper reframes recurrent VAEs through the lens of approximate stationarity and time-shift equivariance, arguing that many real-world sensor signals are best modeled by emphasizing relative temporal dynamics rather than absolute time indices. The authors further connect this inductive bias to the Echo State Property (ESP), using it as a diagnostic tool to analyze state forgetting and the balance between stability and long-term dependency retention.

Empirically, the model is evaluated on five benchmark datasets spanning highly quasi-periodic (sine, ECG, electric motor) to more irregular sensor data. Across multiple metrics—including ELBO, Context-FID, discriminative scores, and latent-space visualizations—RVAE-ST consistently matches or outperforms strong baselines, particularly at long sequence lengths where many competing models degrade.

Key strengths
A clear and well-motivated inductive bias aligned with quasi-periodic and approximately stationary time series.
A simple yet effective progressive training strategy that substantially improves stability and performance on long sequences.
Comprehensive experimental evaluation across diverse datasets and architectural families.
Insightful use of ESP as an analytical lens rather than a purely theoretical claim.

Potential weaknesses
The progressive training schedule is heuristic and dataset-dependent, with limited guidance on principled selection.
Gains are most pronounced for quasi-periodic data, while advantages on more irregular datasets are sometimes marginal.

**Audience:**

Yes

**Audience Explanation:**

The paper is likely to be of interest to a broad segment of the TMLR audience working on representation learning, generative modeling, and sequential data. In particular, researchers concerned with long-horizon time series, inductive biases in neural architectures, or stability issues in recurrent models will find the proposed approach and analysis valuable.

Beyond the specific model, the paper contributes a conceptual framing—approximate equivariance and ESP-based diagnostics—that is applicable beyond VAEs and could inform future work on recurrent diffusion models or other generative frameworks. Its emphasis on simplicity and computational efficiency further increases its practical relevance.

**Claims And Evidence:**

Yes

**Claims Explanation:**

The paper’s central claims—namely that approximate time-shift equivariance combined with progressive sequence-length training improves recurrent generative modeling of long, quasi-periodic time series—are well supported by both empirical and analytical evidence.

The experimental section is thorough and carefully designed. Performance is evaluated using complementary metrics (ELBO, Context-FID, discriminative score, PCA/t-SNE), reducing reliance on any single measure. The ablation study comparing conventional training with the proposed progressive scheme is particularly convincing, showing consistent and statistically significant improvements across all datasets.

In addition, the ESP analysis provides a clear mechanistic explanation for the observed behavior, demonstrating how training balances state forgetting with preservation of meaningful temporal structure. Importantly, the authors avoid overclaiming: they explicitly acknowledge that ESP is used diagnostically rather than as a formal guarantee, which strengthens the credibility of the analysis.

Overall, the evidence is accurate, clearly presented, and appropriately scoped to the claims made.

**Requested Changes:**

1. Provide clearer guidance or heuristics for choosing the initial sequence length and increment schedule in the progressive training scheme, potentially supported by additional ablations or sensitivity analysis.
2. Clarify and streamline the discussion of equivariance in baseline models to better distinguish architectural properties from empirical behavior.
3. Expand discussion on failure modes or scenarios where the inductive bias may be mismatched (e.g., strongly non-stationary or event-driven time series).
4. Consider reporting wall-clock training time or computational cost comparisons to further support claims of efficiency.

---

> ### Author Response · Authors · 2026-01-26
> **Answer to 5aBU**
>
> We thank the reviewer for the constructive feedback and address all points below:
>
> (1) (Progressive training schedule). We have addressed the sensitivity analysis in our response to Reviewer jERd (Requested Change 4). To summarize: performance is robust for small-to-moderate increments, with the +150 schedule achieving the best overall results. Large step sizes (Δl ≥ 300) degrade performance, and training directly at l=1000 without progressive training performs significantly worse.
>
> Regarding the initial sequence length: based on our experience across multiple datasets, starting at l=50 or l=100 both work well. The ablations in Table 5 focus on the increment size rather than the starting point, but our empirical observations suggest that any sufficiently short starting length yields good results. We have added explicit guidance to Section 4.4, recommending to start at l=50 to l=100 and to use moderate increments (Δl ≤ 150).
>
>
> (2) We streamlined the equivariance discussion by explicitly distinguishing architectural enforcement from empirical behavior: the baselines do not enforce time-shift equivariance by design, whereas RVAE-ST does (Appendix A.2 Fig. 6). We added a short transition and clarified wording accordingly.
>
>
> (3) We addressed this by adding a dedicated “Limitations” subsection in the discussion. It explicitly outlines scenarios where our inductive bias is mismatched, including strongly non-stationary and event-driven series (trends, regime shifts, structural breaks), small-data regimes, low temporal resolution (e.g., ETTm2), and signals with abrupt discontinuities (illustrated on MetroPT3 via frequent sharp drops).
>
>
> (4) (Computational cost). Thank you for this suggestion. We have added a wall-clock training time comparison to the revised manuscript (Appendix A.1, Table 4).
>
> On the Electric Motor dataset at l=1000, RVAE-ST with subsequent training requires approximately 8 hours, compared to 11 minutes for TimeVAE, 40 minutes for WaveGAN, 1h20min for Time-Transformer, and 2.5 hours (+1 hour sampling time) for Diffusion-TS. TimeGAN requires approximately 20 days.
>
> The convolution-based models (TimeVAE, WaveGAN) are fastest, while recurrent models require longer training times. Diffusion-TS additionally require substantial time for sample generation after training. We note that these measurements were performed on different hardware and should be interpreted as order-of-magnitude estimates.

---

### Review · Reviewer_jerd · 2026-01-04

**Summary Of Contributions:**

The paper proposes a generative model for long, quasi-periodic time series called RVAE-ST (Recurrent Variational Autoencoder with Subsequent Training). The authors argue that while Transformers are currently popular, they lack the necessary inductive bias (time-shift equivariance) for modeling quasi-periodic sensor data (e.g., motors, ECGs).

The main contributions are:

1. Architecture: A recurrent VAE topology that employs a "repeat-vector" mechanism and time-distributed output layers to enforce parameter independence from sequence length and approximate time-shift equivariance.
2. Training Scheme: A "Subsequent Training" curriculum that progressively increases sequence length during training (e.g., from 100 to 1000) to mitigate the optimization instability of LSTMs on long horizons .
3. Theoretical Analysis: An application of the Echo State Property (ESP) as a diagnostic tool to demonstrate that the trained model effectively balances forgetting initialization artifacts with retaining input-driven structure.
4. Empirical Evaluation: Comparisons against five baselines (including TimeGAN, TimeVAE, and Diffusion-TS) across five datasets, showing superior performance on periodic data using Context-FID and Discriminative scores.

## Strength

1. The hypothesis regarding inductive biases (recurrence vs. positional encoding) is well-motivated and theoretically grounded.
2. The progressive training scheme effectively addresses the well-known vanishing gradient problem in LSTMs for this specific task.
3. The analysis of the Echo State Property provides interesting insights into how RNNs model stationarity.

## Areas for Improvement:

1. Baseline Configuration: The comparison would benefit from ensuring baselines are tuned to the same degree as the proposed method to fully validate the performance gains .
2. Implementation Details: The specific configuration used for WaveGAN (training on very long sequences) may unintentionally handicap its performance.
3. Metric Interpretation: There should be a clarification regarding discrepancies between the quantitative metrics (Discriminative Score) and the visual quality of the samples.

**Audience:**

Yes

**Audience Explanation:**

The paper tackles a relevant and practical problem of generating long-horizon sensor data.

1. The argument for revisiting RNNs for specific data types (quasi-periodic) challenges the current dominance of Transformers and contributes valuable nuance to the field's understanding of inductive biases.
2. The progressive training curriculum is a generalizable technique that could be very useful to practitioners working with recurrent models in other domains .

**Broader Impact Concerns:**

No concerns. The work is methodological and focuses on sensor data generation, which presents minimal ethical risk.

**Claims And Evidence:**

No

**Claims Explanation:**

The paper presents a compelling hypothesis, but the current empirical evidence does not yet fully support the claim of superiority over state-of-the-art baselines. I believe addressing the following points would make the evidence much more convincing:

1. Baseline Hyperparameters: The authors note that they did not perform extensive hyperparameter tuning and used the default hyperparameters across models to ensure comparability . However, different architectures (e.g., GANs vs. Diffusion) often require very different settings to converge optimally. Without basic tuning of the baselines, it is difficult to determine if the performance gap is due to the proposed architecture or simply suboptimal baseline configurations.

2. WaveGAN Implementation: The choice to train WaveGAN on sequences of length 2^14 (16,384) due to implementation constraints , while the target task is only length 1000, likely places this baseline at a severe disadvantage. This significantly increases the complexity of the learning task for WaveGAN compared to the other models.

3. Metric Reliability: There appears to be a contradiction between the metrics and the visuals. For example, on the Sine dataset, Diffusion-TS achieves a significantly better Discriminative Score (0.021) than the proposed RVAE-ST (0.122), yet the visual samples in Figure 5 suggest RVAE-ST is far superior . This raises questions about whether the Discriminative Score is a reliable proxy for quality in this specific context.

**Requested Changes:**

I believe the following adjustments would significantly strengthen the paper and ensure the claims are robust:

1. Clarification on Baselines (Critical): To remove any doubt regarding the superiority of RVAE-ST, could the authors please perform a basic hyperparameter search (e.g., learning rate, latent dimension) for the key baselines (TimeGAN, Diffusion-TS)? Demonstrating that RVAE-ST outperforms even tuned baselines would make their claim stronger.

2. Refining the WaveGAN Comparison (Critical): I strongly suggest adapting the WaveGAN implementation to support a sequence length closer to the target (e.g., 1024) rather than 16384. If this is not feasible, please add a discussion or ablation clarifying how this discrepancy might impact the results. This ensures the baseline represents the method's potential rather than a limitation of a specific codebase.

3. Insight into Metrics (Critical): Could the authors please elaborate on why Diffusion-TS achieves a superior Discriminative Score on the Sine dataset despite the noisy visuals?. A brief discussion on the limitations of this metric for this specific task would help readers interpret the results more accurately.

4. Curriculum Sensitivity (Strengthening): The "Subsequent Training" scheme is a valuable contribution, but the current step size (100) and start length (100) are heuristic. To demonstrate robustness, could you provide a sensitivity analysis? Specifically, does the method still work effectively if the step size is larger (e.g., +200 or +500)? This would confirm if the curriculum is a generalizable method or requires precise tuning.

5. Architectural Inductive Bias (Strengthening): The paper claims the performance gain comes from the specific "approximate equivariant" architecture (Repeat Vector + Time-Distributed decoder). To isolate this benefit, could you perform an ablation comparing your proposed decoder against a standard LSTM-VAE decoder (where z initializes the hidden state)? This would verify that the improvement is indeed due to the specific inductive bias and not just the use of LSTMs.

---

> ### Author Response · Authors · 2026-01-26
> **Answer to jerd**
>
> We thank the reviewer for the constructive feedback and address all points below:
>
> (1) **(Baseline hyperparameters / tuning).** Thank you for the comment. Our statement about "not performing extensive hyperparameter tuning" was imprecise. We will revise it in the manuscript.
>
> To clarify our experimental protocol: **RVAE-ST uses a single, fixed configuration across all datasets and all sequence lengths.** For the baselines, we adopt official configurations from the respective repositories. For Diffusion-TS, we use the authors' Sine-specific config for the Sine dataset. For all other datasets, including ETTM, we use the authors' ETTH config, which has more layers than the Sine config. Note that ETTH and ETTM share the same configuration and differ only in temporal resolution.See Appendix Hyperparameters and model configs for full details.
>
> For TimeGAN, we did attempt additional tuning given its poor performance on long sequences. However, with training times of approximately three weeks on an NVIDIA DGX A100 for l=1000, systematic hyperparameter exploration was not feasible. This itself highlights the scalability limitations of this architecture.
>
> We would like to emphasize that our claim is not to achieve globally tuned SOTA performance, but to demonstrate that our inductive bias and training scheme provide a robust approach to long-horizon time series generation, working reliably across multiple datasets without requiring dataset-specific tuning.
>
> (2) **(WaveGAN length mismatch).**
> We refactored the upsampling (generator) and downsampling (discriminator) paths in WaveGAN to support a reduced sequence length of 1024 (instead of the original 16384). The resulting FID scores (lower=better) were substantially worse than the original setup (trained at 2¹⁴ and then split):
>
> |WaveGAN setup|l=100|l=300|l=500|l=1000|
> |---|---|---|---|---|
> |trained at 2^{14}, then split|0.55 ± 0.04|0.75 ± 0.07|0.87 ± 0.14|1.41 ± 0.24|
> |trained at 1024, then split|2.29 ± 0.15|3.05 ± 0.25|3.35 ± 0.30|3.02 ± 0.31|
>
> This shows that WaveGAN’s architecture and capacity are tuned for long-form generation (e.g., 2^{14}+ samples) and do not transfer well to shorter sequences like 1024. Adapting it to perform competitively at that scale would effectively require us to design a new model. Since the original variant performs better, we retain it in the paper.
>
> As noted in the manuscript, WaveGAN is not central to our claims. It serves as a supplemental reference for inductive bias comparison. We already describe its architectural constraints and clarify that the reported results reflect a faithful use of the original implementation, not a re-tuned or length-optimized variant.
> Interestingly, in our setting, WaveGAN consistently outperforms TimeGAN—contrary to the findings reported in the original TimeGAN paper. This highlights that such relative rankings can depend strongly on sequence length  data characteristics.

---

> > ### Author Response · Authors · 2026-01-26
> > **Answer to jerd 2**
> >
> > (3) **(Discriminative Score vs. visual quality on Sine).** We believe there is a misunderstanding regarding Table 2. On Sine at l=1000, Table 2 reports **0.021 for RVAE-ST** and **0.428 for Diffusion-TS**. The value 0.122 mentioned by the reviewer corresponds to RVAE-ST on ETT, not Sine. With the correct mapping, there is no contradiction: the qualitative samples in Figure 5 are fully consistent with the discriminative score results, where RVAE-ST performs substantially better than Diffusion-TS on Sine.
> >
> > (4) **(Curriculum Sensitivity).** Thank you for this suggestion. We have conducted a sensitivity analysis and added it to the revised manuscript (A.1 Training Scheme Ablations.)
> >
> > We evaluated several sequence-length schedules on the Electric Motor and Sine datasets, ranging from fine increments (+50) to coarse increments (+450). The results show that subsequent training is robust across a range of step sizes. Moderate increments (e.g., +50, +100, +150) all perform similarly well, with the +150 schedule achieving the best overall performance across both datasets. Performance degrades when the step size becomes large (roughly Δl ≥ 300), most notably on Sine.
> >
> > Finally, training directly at l=1000 without subsequent training performs significantly worse on both datasets, which underscores that the progressive scheme is critical for successful learning at long horizons.
> >
> > (5) **(Architectural Inductive Bias).** Thank you for this suggestion. We have conducted the requested ablation and added it to the revised manuscript (A.3 Ablation: Decoder inductive bias).
> >
> > We compare RVAE-ST against a control decoder that keeps the same recurrent backbone (four-layer LSTM stack) but removes the key constraints of our design: length-independent parameterization and approximate time-shift equivariance. The control decoder maps z through a dense layer to a length-dependent representation and uses a global output projection that can implement position-specific mappings.
> >
> > The results confirm that the inductive bias of our decoder is critical for long-horizon performance. While the control decoder achieves comparable results on short sequences (e.g., on ECG at l=100), its performance degrades substantially as sequence length increases. This is most pronounced on Sine and ETT at l=1000, where FID scores increase by an order of magnitude compared to RVAE-ST. In contrast, RVAE-ST remains stable across all sequence lengths.
> >
> > These results demonstrate that the performance gains are not simply due to using LSTMs, but specifically due to the repeat-vector and time-distributed output structure that enforces our intended inductive bias.

---

### Review · Reviewer_8g3S · 2026-01-07

**Summary Of Contributions:**

The authors address the problem of training generative models for time series which may exhibit periodic, or approximately periodic, behavior. One of the main challenges of this problem is both the length and the amount of series required to effectively capture this structure, and the inductive biases that may help or hinder the ability to capture this type of temporal structure. The authors propose a Recurrent Variational Autoencoder Subsequent Train (RVAE-ST) method, which combine specific choices of inductive biases, network topologies, and a training scheme, to address this problem. To illustrate the effectiveness of their approach, they perform experiments on five benchmark datasets, and report both qualitative and quantitative metrics.

**Additional Comments:**

There are no additional comments.

**Audience:**

Yes

**Audience Explanation:**

I believe that the problem of modeling quasi-periodic structures is relevant both to the readers of TMLR and beyond. As the authors correctly point out, there are medical applications where the _deviation_ from a periodic behavior is of clinical interest. Although generating such time series does not seem, to my knowledge, to have a direct application, being able to successfully generate them suggests that one has identified a proper set of parameters that describe a quasi-periodic structure.

**Broader Impact Concerns:**

There are no broader impact concerns.

**Claims And Evidence:**

No

**Claims Explanation:**

One of my main concerns is the property of "quasi-periodicity." Although the concept is fairly intuitive, the authors do not provide a proper, working mathematical definition for it, and thus it is difficult to assess to which degree the proposed architecture captures "quasi-periodicity." The authors do perform comparisons between different methods in relevant datasets. However, these comparisons use metrics that may or may not reflect the successful approximation of quasi-periodicity. For this reason, I believe that it is necessary to propose a metric to assess "quasi-periodicity" to justify the claims of the paper.

To further explain my point, one could ask whether there are _degrees_ of quasi-periodicity. If there are, one could quantify what is the degree of quasi-periodicity exhibited in each dataset, and thus determine if some of them are more challenging to analyze. Another aspect is the relation between quasi-periodic structures and sequence length: when do quasi-periodic structures become noise-like as the sequence length increases? Finally, some quasi-periodic structures can be an artifact of sequence length. For instance, consider the function

$$
	u(t) = \cos(2\pi f_0 t + \epsilon \cos(2\pi f_1 t))
$$

for $\epsilon$ small. Its instantaneous frequency is

$$
	f_0 - 2\pi \epsilon \sin(2\pi f_1 t)
$$

and therefore there are _periodic variations_ of the frequency around $f_0$. If one observed the time series over an interval of length smaller than $1/f_1$ then this periodic variation could plausibly be interpreted as quasi-periodicity. However, if it is observed over an interval of length much larger than $1/f_1$, then it becomes a periodic structure.

As an aside, the concepts of _quasiperiodic functions_ and _almost periodic functions_ do exists in the mathematical literature. Although their definition may be too abstract, perhaps they could be helpful to define suitable quantitative metrics to determine a degree of "quasi-periodic-ness."

**Requested Changes:**

My requests are as follows.

1. Perhaps my main request is to provide a workable definition for quasi-periodicity that enables principled comparisons between methods. Otherwise it is difficult to assess which structural features are being learned.
2. Although the authors perform extensive experiments in relevant datasets, there is one simple synthetic experiment that perhaps could illustrate the behavior of the architecture: generating synthetic time series with prescribed Power Spectral Density (PSD). Although this type of process generates statistically stationary time series, a typical sample path may exhibit quasi-periodic structures. It would be interesting to see if the architecture, depending on sequence length, learns to generate time series with the same PSD or if it hallucinates quasi-periodic structures.

---

> ### Author Response · Authors · 2026-01-26
> **Answer to 8g3S**
>
> Thank you for raising the terminology issue. We agree that the term *quasi-periodic* is used differently across communities. In classical mathematics/dynamical systems and harmonic analysis, *quasi-periodic* often refers to structured signals generated by a finite number of incommensurate frequencies (and is therefore not simply “noisy periodicity”). In contrast, in applied time-series anomaly detection, *quasi-periodic* is frequently used as an umbrella term for *repetitive motifs that recur with imperfect regularity*—e.g., variable cycle lengths, phase jitter, drift, noise, missing cycles, and occasional anomalies. This applied usage appears widely in the anomaly-detection literature on “quasi-periodic time series” \[R2,R3,R4\] and is also explicitly articulated in recent industrial anomaly-detection discussions \[R1\].
>
> Our paper uses *quasi-periodic* strictly in this applied anomaly-detection sense. Concretely, we target time series that exhibit a repeating pattern but where strict periodicity fails due to realistic effects such as timing errors, drift, noise, and anomalies. This is consistent with the definition-by-description used in \[R1\], and with the way the term is operationalized in anomaly-detection pipelines that rely on segmentation/alignment of approximately repeating cycles \[R2,R4\] or compare algorithms on quasi-periodic consumption traces \[R3\].
>
> To address the reviewer’s concern and avoid confusion with the classical meaning, we will revise the manuscript as follows:
>
> 1. **Explicit terminology clarification (added to the paper).** We will add a short “Terminology” paragraph early in the paper stating that we use *quasi-periodic* in the applied anomaly-detection sense (imperfectly repeating motifs), consistent with \[R1–R4\], and that this differs from the classical incommensurate-frequency notion.
>
> 2. **Operational characterization (added to the paper; not a new formal definition).** Our goal is not to introduce a new mathematical definition of “closeness to periodicity,” but to introduce an *inductive bias* for learning in the quasi-periodic anomaly-detection regime studied in \[R2–R4\]. Operationally, the data regime we consider can be summarized as:
>
>    > A time series that admits a segmentation into cycles such that consecutive cycles are similar after mild alignment (e.g., small time-warping/phase shift) and normalization, while residual components capture drift/noise/anomalies.
>
> This perspective is closely aligned with how quasi-/pseudo-periodic streams are handled in the data-stream and segmentation literature, where deviations from strict periodicity are treated as variations around a repeating pattern rather than as violations that invalidate cycle structure \[R5,R6\].
>
> 3. **Connection to our method (clarified in the paper).** The motivation for our approach is precisely this “imperfect repetition” regime:
>
>    - **Approximate time-shift equivariance** is intended to encourage representations that are stable under phase shifts and cycle-to-cycle timing variability, matching the practical needs of quasi-periodic anomaly detection \[R2,R4\].
>
>    - **Progressive training** is designed to stabilize learning on long horizons where repetition exists but is not exact (due to drift/noise/anomalies), which is characteristic of the quasi-periodic benchmarks studied in \[R3\] and the broader industrial framing in \[R1\].
>
> Finally, we acknowledge that many formal notions exist for relaxing periodicity (e.g., almost-periodic concepts), but no single formalism cleanly covers the heterogeneous distortions emphasized in applied anomaly-detection settings \[R1–R4\]. Accordingly, we emphasize empirical validation in the same spirit as prior anomaly-detection work on quasi-periodic time series \[R2–R4\], and we will ensure the revised manuscript clearly states the intended usage of the term.

---

> ### Author Response · Authors · 2026-01-26
> **Answer to 8g3S 2**
>
> ### References
>
> [R1] arXiv:2506.16815. (Recent industrial/anomaly-detection discussion describing quasi-periodic time series as repetitive patterns with variable periods/segment lengths due to timing errors.)
>
> [R2] Liu, Fan, et al. "Anomaly detection in quasi-periodic time series based on automatic data segmentation and attentional LSTM-CNN." *IEEE Transactions on Knowledge and Data Engineering*, 2020.
>
> [R3] Zangrando, Niccolò, et al. "Anomaly detection in quasi-periodic energy consumption data series: a comparison of algorithms." *Energy Informatics*, 2022.
>
> [R4] Tang, Xiaolan, et al. "An automatic segmentation framework of quasi-periodic time series through graph structure." *Applied Intelligence*, 2023.
>
> [R5] Tang, Lv-an, et al. "Effective variation management for pseudo periodical streams." In: *Proceedings of the 2007 ACM SIGMOD International Conference on Management of Data*, 2007.
>
> [R6] Yin, Ning, et al. "A segment-wise method for pseudo periodic time series prediction." In: *International Conference on Advanced Data Mining and Applications*, 2014.

---

> ### Author Response · Authors · 2026-01-26
> **Answer to 8g3S Request (2)**
>
> We thank the reviewer for this suggestion. Following your advice, **we added a PSD experiment in Appendix A.4 with PSD comparisons and example samples (Figures 11 and 12).**
>
> We generated 30,000 sequences from a target PSD with two Gaussian peaks at f_1 = 0.08 and f_2 = 0.12 (normalized frequency) and trained RVAE-ST using our subsequent training scheme up to l = 1000.
>
> The results show that RVAE-ST captures both dominant frequencies up to l = 900, though it acts as a filter that suppresses the noise floor and spectral content between the peaks. At l = 1000, f_2 becomes attenuated relative to f_1. Importantly, we observe no hallucinated quasi-periodic structures. The generated samples match the originals well visually, but show increasing amplitude compression at longer sequence lengths.
>
> In short: RVAE-ST learns the dominant frequency structure rather than hallucinating patterns, though it produces slightly "cleaner" outputs by filtering weaker spectral components—an effect that becomes more pronounced with increasing sequence length.

---

> ### Comment · Reviewer_8g3S · 2026-02-06
> **Reply**
>
> I thank the authors for this detailed response and the references in their follow up reply. This addresses the main point of my review. The paragraph in the introduction now clearly states the definition of "quasi-periodicity" that the authors use throughout the manuscript, and it provides suitable references to the reader.
>
> The experiments in Appendix A.4 now provide a clear example of the behavior of the proposed architecture when trained in synthetic stationary series.

---

### Author Response · Authors · 2026-01-26
**Global Response to All Reviewers**

We thank all reviewers for their constructive feedback and thoughtful engagement with our work. The reviews have helped us strengthen the manuscript in several important ways.

In response to the comments, we have made the following key revisions:

• We clarified our use of "quasi-periodic" throughout the manuscript, particularly in the Introduction and Section 4.1, distinguishing the applied anomaly-detection sense from the classical mathematical definition (Reviewer 8g3S).

• We conducted a PSD experiment (Appendix A.4), a training schedule sensitivity analysis (Appendix A.1), and a decoder ablation study (Appendix A.3) as requested.

• We added wall-clock training time comparisons (Appendix, Table 6).

• We added a dedicated subsection discussing failure modes and scenarios where our inductive bias may be mismatched (6.1 Limitations).

We address each reviewer's specific points in detail below.

---

### Decision · Action_Editor_AuHZ · 2026-02-18

**Recommendation:** Accept with minor revision

**Additional Comments:**

All three reviewers feel positive about the paper and appreciate its contributions, including approximate time-shift equivariance, sequence-length–independent parameterization, and progressive training. Thorough evaluation and promising results were highlighted as well. The concerns raised in the reviews, in particular the imprecise definition of 'quasi-periodic' and insufficient baseline optimization, were adequately addressed by the authors in their response. Additional experiments and ablation studies requested by the reviewers, such as a step size sensitivity analysis, PSD analysis, and training time comparison further strengthened the paper, as did the inclusion of failure cases.

While I agree with the reviewers on these positive aspects, I would like to ask the authors to improve the following points:

- Relative to its complexity and contributions, the paper is very long. I do not want to prescribe a page count but urge the authors to improve the information density of the presentation.
- The name of the proposed approach (RVAE-ST) is not a good choice, because it does not reflect the equivariance property of the architecture, which is a key contribution.
- Figure 2 is a key figure but lacks important information about the structure of the stacked LSTM and time-distributed layer. Please rework the figure so that it illustrates these concepts more clearly.

Once these changes have been made, the paper will be recommended for acceptance.

**Audience:**

Yes

**Audience Explanation:**

The paper explores the underrepresented task of modeling quasi-periodic time-series and proposes two solutions (approximate time-shift equivariance and progressive training) that could be of more general interest, e.g., in the context of data with long-range dependencies and architectures with inductive biases. The RVAE-based framework holds up surprisingly well compared to modern approaches based on transformers or diffusion, which could be a two-way incentive: revisiting traditional model families in specific settings or porting promising insights to more recent architectures.

**Claims And Evidence:**

Yes

**Claims Explanation:**

The paper proposes a type of recurrent VAE in which inductive biases at the architectural level ensure approximate time-shift equivariance. This property is particularly useful in the context of quasi-periodic time series, where it allows the model to learn local patterns independent of their absolute position in a temporal sequence.

At a theoretical level, the proposed architecture is well motivated and has an intuitive appeal. At an empirical level, the provided analysis based on the Echo State Property (ESP) - even if an imperfect proxy - is aligned with the claim of approximate time-shift equivariance. An ablation study comparing the proposed method against an RVAE variant without approximate time-shift equivariance shows strong performance degradation of the baseline and validates the effectiveness of the key design choices (repeat vector and time-distributed linear layer). The paper's second major contribution, a progressive training scheme, is shown to lead to significant performance gains over a baseline with standard training.

Comparisons to five baselines on five datasets exhibiting varying degrees of quasi-periodic behaviour show promising performance w.r.t. Context-FID scores and Discriminative scores, especially for longer sequences.

---

> ### Author Response · Authors · 2026-02-21
> **Thank you for the decision**
>
> Dear Action Editor and Reviewers,
> Thank you very much for the positive decision on our paper. We are pleased that our revisions adequately addressed the reviewers' concerns.
> We will carefully incorporate the Action Editor's feedback into the camera ready version of the manuscript and submit it within the given timeframe.

---

> ### Author Response · Authors · 2026-03-16
> **Camera Ready Revision**
>
> Dear Action Editor,
>
> thank you for your constructive feedback. We have addressed all three points raised in your recommendation. Overall, these changes reduced the paper from 46 to 37 pages, including a reduction of 2 pages in the main text.
>
> **1. Model Renaming:** We renamed our approach from RVAE-ST to AEQ-RVAE-ST (Approximate Equivariant Quasi-periodic RVAE - subsequent train) to explicitly reflect the equivariance property in the model name.
>
> **2. Figure 2:** We reworked Figure 2 to more clearly illustrate the structure of the stacked LSTM, the time-distributed layer, and the weight sharing mechanism. Additionally, we added a reference to the appendix where the input and output dimensions of the decoder are explicitly listed.
>
> **3. Improved Information Density:** We made substantial efforts to reduce the paper's length while preserving all relevant content. Specifically:
>
> In the main text, we removed the Prerequisites section and integrated the necessary information into the Methods section (Section 3.3). We reformulated the dataset descriptions more compactly, shortened the first figure's caption, and reduced the ESP discussion section by removing enumerated lists in favor of flowing text and eliminating redundancies between the section text and the caption.
>
> In the appendix, we condensed several subsections for brevity. For the PCA and t-SNE plots, we removed the intermediate sequence lengths (l=300 and l=500) from all plots and combined PCA and t-SNE visualizations into single figures per dataset, reducing the total number of figures. We also removed redundant appendix plots for datasets already shown in the main text (ECG and EM for PCA).
>
> We believe these changes substantially improve the information density of the paper as requested.
>
> Best regards,